# PAC-Bayes-Chernoff bounds for unbounded losses

**Ioar Casado**
Machine Learning Group
Basque Center for Applied Mathematics (BCAM)
`icasado@bcamath.org`

**Luis A. Ortega**
Machine Learning Group
Computer Science Dept. - EPS.
Universidad Autónoma de Madrid
`luis.ortega@uam.es`

**Aritz Pérez**
Machine Learning Group
Basque Center for Applied Mathematics (BCAM)
`aperez@bcamath.org`

**Andrés R. Masegosa**
Department of Computer Science
Aalborg University
`arma@cs.aau.dk`

## Abstract

We introduce a new PAC-Bayes oracle bound for unbounded losses that extends Cramér-Chernoff bounds to the PAC-Bayesian setting. The proof technique relies on controlling the tails of certain random variables involving the Cramér transform of the loss. Our approach naturally leverages properties of Cramér-Chernoff bounds, such as exact optimization of the free parameter in many PAC-Bayes bounds. We highlight several applications of the main theorem. Firstly, we show that our bound recovers and generalizes previous results. Additionally, our approach allows working with richer assumptions that result in more informative and potentially tighter bounds. In this direction, we provide a general bound under a new *model-dependent* assumption from which we obtain bounds based on parameter norms and log-Sobolev inequalities. Notably, many of these bounds can be minimized to obtain distributions beyond the Gibbs posterior and provide novel theoretical coverage to existing regularization techniques.

## 1 Introduction

PAC-Bayes theory provides powerful tools to analyze the generalization performance of randomized learning algorithms —for an introduction to the subject see the recent surveys of Guedj (2019); Alquier (2024) and Hellström et al. (2023)—. Let $\boldsymbol{\Theta}$ be a model class parametrized by $\boldsymbol{\theta} \in \mathbb{R}^d$. With a small abuse of notation, $\boldsymbol{\theta} \in \boldsymbol{\Theta}$ will represent both a model and its parameters. Instead of learning a single model, we consider the set of probability measures over our class, $\mathcal{M}_1(\boldsymbol{\Theta})$, and aim to find the optimal distribution $\rho^* \in \mathcal{M}_1(\boldsymbol{\Theta})$. This approach is more robust than finding the single best model in $\boldsymbol{\Theta}$, and is tightly related to Bayesian and ensemble methods. The learning algorithm infers this distribution from a sequence of $n$ training data points $D = \{\boldsymbol{x}_i\}_{i=1}^n$, which are assumed to be i.i.d. sampled from an unknown distribution $\nu(\boldsymbol{x})$ with support in $\mathcal{X} \subseteq \mathbb{R}^k$.

Given a *loss function* $\ell : \boldsymbol{\Theta} \times \mathcal{X} \to \mathbb{R}_+$, bounding the gap between the *population risk* $L(\boldsymbol{\theta}) := \mathbb{E}_\nu[\ell(\boldsymbol{\theta}, X)]$ and the *empirical risk* $\hat{L}(\boldsymbol{\theta}, D) := \frac{1}{n} \sum_{i=1}^n \ell(\boldsymbol{\theta}, \boldsymbol{x}_i)$ of individual models is the standard approach in statistical learning theory. In contrast, PAC-Bayes theory provides high-probability bounds over the *population Gibbs risk* $\mathbb{E}_\rho[L(\boldsymbol{\theta})]$ in terms of the *empirical Gibbs risk* $\mathbb{E}_\rho[\hat{L}(\boldsymbol{\theta}, D)]$ and an extra term measuring the dependence of $\rho$ to the dataset $D$. This second term involves an information measure —usually the Kullback-Leibler divergence $KL(\rho|\pi)$— between the data

38th Conference on Neural Information Processing Systems (NeurIPS 2024).

dependent *posterior*[1] $\rho \in \mathcal{M}_1(\boldsymbol{\Theta})$ and a *prior* $\pi \in \mathcal{M}_1(\boldsymbol{\Theta})$, chosen before observing the data. These bounds hold simultaneously for every $\rho \in \mathcal{M}_1(\boldsymbol{\Theta})$, hence minimizing them with respect to $\rho$ provides an appealing approach to derive new learning algorithms with theoretically sound guarantees.

The foundational papers on PAC-Bayes theory (Shawe-Taylor and Williamson, 1997; McAllester, 1998, 1999; Seeger, 2002) worked with classification problems under bounded losses, usually the zero-one loss. This framework was significantly extended in Catoni (2007), who introduced some of the first bounds for unbounded losses. McAllester's bound (McAllester, 2003) is one of the most representative results for bounded losses: for any $\pi \in \mathcal{M}_1(\boldsymbol{\Theta})$ independent of $D$ and every $\delta \in (0, 1)$, we have

$$\mathbb{E}_\rho[L(\boldsymbol{\theta})] \leq \mathbb{E}_\rho[\hat{L}(D, \boldsymbol{\theta})] + \sqrt{\frac{KL(\rho|\pi) + \log\frac{2\sqrt{n}}{\delta}}{2n}} , \tag{1}$$

simultaneously for every $\rho \in \mathcal{M}_1(\boldsymbol{\Theta})$, where the above inequality holds with probability no less than $1 - \delta$ over the choice of $D \sim \nu^n$. Another significant example under the bounded loss assumption is the Langford-Seeger-Maurer bound —after (Langford and Seeger, 2001; Seeger, 2002; Maurer, 2004)—. Under the same conditions as above,

$$kl\left(\mathbb{E}_\rho[\hat{L}(D, \boldsymbol{\theta})], \mathbb{E}_\rho[L(\boldsymbol{\theta})]\right) \leq \frac{KL(\rho|\pi) + \log\frac{2\sqrt{n}}{\delta}}{n} , \tag{2}$$

where $kl$ is the so-called binary-kl distance, defined as $kl(a, b) := a \ln \frac{a}{b} + (1 - a) \ln \frac{1-a}{1-b}$.

These bounds illustrate typical trade-offs in PAC-Bayes theory. In (1), the relation between the empirical and the population risk is easy to interpret because the expected loss is bounded by the empirical loss plus a complexity term. More crucially, the right-hand side of the bound can be directly minimized with respect to the posterior $\rho$, resulting in the Gibbs posterior (Guedj, 2019). Conversely, (2) is known to be tighter than (1), but it is not straightforward to minimize because it requires inverting $kl\left(\mathbb{E}_\rho[\hat{L}(D, \boldsymbol{\theta})], \cdot\right)$ —several techniques deal with this issue (Thiemann et al., 2017; Reeb et al., 2018)—.

With the trade-offs among explainability, tightness, and generality in mind, the PAC-Bayes community has come up with novel bounds with applications in virtually every area of machine learning, ranging from the study of particular algorithms —linear regression (Alquier and Lounici, 2011; Germain et al., 2016), matrix factorization (Alquier and Guedj, 2017), kernel PCA (Haddouche et al., 2020), ensembles (Masegosa et al., 2020; Wu et al., 2021; Ortega et al., 2022) or Bayesian inference (Germain et al., 2016; Masegosa, 2020)— and generic versions of PAC-Bayes theorems (Bégin et al., 2016; Rivasplata et al., 2020) to the study of the generalization of deep neural networks (Dziugaite and Roy, 2017; Rivasplata et al., 2019).

Such applications often require relaxing the assumptions of the classical PAC-Bayesian framework, such as data-independent priors, i.i.d. data or bounded losses. Here we are interested in the case of PAC-Bayes bounds for unbounded losses —such as the squared error loss and the log-loss—.

**PAC-Bayes bounds for unbounded losses**

The main challenge of working with unbounded losses is that one needs to deal with an exponential moment term which cannot be easily bounded without specific assumptions about the tails of the loss. The following theorems, fundamental starting points for many works on PAC-Bayes theory over unbounded losses, illustrate this point.

**Theorem 1** ((Alquier et al., 2016; Germain et al., 2016)). *Let $\pi \in \mathcal{M}_1(\boldsymbol{\Theta})$ be any prior independent of $D$. Then, for any $\delta \in (0, 1)$ and any $\lambda > 0$, with probability at least $1 - \delta$ over draws of $D \sim \nu^n$,*

$$\mathbb{E}_\rho[L(\boldsymbol{\theta})] \leq \mathbb{E}_\rho[\hat{L}(D, \boldsymbol{\theta})] + \frac{1}{\lambda n}\left[KL(\rho|\pi) + \log\frac{f_{\pi,\nu}(\lambda)}{\delta}\right] ,$$

*simultaneously for every $\rho \in \mathcal{M}_1(\boldsymbol{\Theta})$. Here $f_{\pi,\nu}(\lambda) := \mathbb{E}_\pi \mathbb{E}_{\nu^n}\left[e^{\lambda n\, (L(\boldsymbol{\theta}) - \hat{L}(D, \boldsymbol{\theta}))}\right]$.*

---

[1] In PAC-Bayes theory the term *posterior* is used to emphasize that $\rho \in \mathcal{M}_1(\boldsymbol{\Theta})$ depends on training data, and does not necessarily refer to the *Bayesian posterior*. We refer the reader to Germain et al. (2016) for a discussion of the relation between PAC-Bayesian and Bayesian methods.

This is an *oracle* bound, because $f_{\pi,\nu}(\lambda)$ depends on the data generating distribution $\nu^n$. To obtain empirical bounds from the theorem above, the exponential term $f_{\pi,\nu}$ is usually bounded by making the appropriate assumptions on the tails of the loss, such as the Hoeffding assumption (Alquier et al., 2016), sub-Gaussianity (Alquier and Guedj, 2018; Xu and Raginsky, 2017), sub-gamma (Germain et al., 2016) or sub-exponential (Catoni, 2004). See Section 5 of Alquier (2024) for an overview.

Many of these assumptions are generalized by the notion that the *cumulant generating function (CGF)* of the (centered) loss, $\Lambda_{\boldsymbol{\theta}}(\lambda)$, exists and is bounded (Banerjee and Montúfar, 2021; Rodríguez-Gálvez et al., 2024). Remember that we say $\Lambda_{\boldsymbol{\theta}}(\lambda)$ exists if it is bounded in some interval $[0, b)$ with $b > 0$.

**Definition 2** (Bounded CGF). A loss function $\ell$ has bounded CGF if for all $\boldsymbol{\theta} \in \boldsymbol{\Theta}$, there is a convex and continuously differentiable function $\psi : [0, b) \to \mathbb{R}_+$ such that $\psi(0) = \psi'(0) = 0$ and

$$\Lambda_{\boldsymbol{\theta}}(\lambda) := \log \mathbb{E}_\nu \left[ e^{\lambda \, (L(\boldsymbol{\theta}) - \ell(\boldsymbol{x}, \boldsymbol{\theta}))} \right] \leq \psi(\lambda) \text{ for all } \lambda \in [0, b). \tag{3}$$

We will say that a loss function $\ell$ is $\psi$-bounded if it satisfies the above assumption under the function $\psi$. In this setup, Banerjee and Montúfar (2021) obtain the following PAC-Bayes bound under the Bounded CGF assumption:

**Theorem 3.** *Let $\ell$ be a loss with $\psi$-bounded CGF and $\pi \in \mathcal{M}_1(\boldsymbol{\Theta})$ any prior independent of $D$. Then, for any $\delta \in (0, 1)$ and any $\lambda \in (0, b)$, with probability at least $1 - \delta$ over draws of $D \sim \nu^n$,*

$$\mathbb{E}_\rho[L(\boldsymbol{\theta})] \leq \mathbb{E}_\rho[\hat{L}(D, \boldsymbol{\theta})] + \frac{KL(\rho|\pi) + \log \frac{1}{\delta}}{\lambda n} + \frac{\psi(\lambda)}{\lambda},$$

*simultaneously for every $\rho \in \mathcal{M}_1(\boldsymbol{\Theta})$.*

Theorems 1 and 3 illustrate a pervasive problem of many PAC-Bayes bounds for unbounded losses: they often depend on a free parameter $\lambda > 0$ —see for example (Alquier et al., 2016; Hellström and Durisi, 2020; Guedj and Pujol, 2021; Banerjee and Montúfar, 2021; Haddouche et al., 2021)—. The choice of this free parameter is crucial for the tightness of the bounds, and it cannot be directly optimized because its choice is prior to the draw of data, while the optimal $\lambda$ would be data-dependent. The standard approach is to optimize $\lambda$ over a grid using union-bound arguments, but the resulting $\lambda$ is not guaranteed to be optimal. Seldin et al. (2012) carefully design the grid so that the optimal $\lambda$ is inside its range, but their work only deals with bounded losses. See the discussion in Section 2.1.4 of Alquier (2024) for an overview.

Recently, Rodríguez-Gálvez et al. (2024) improved the union-bound approach for the bounded CGF scenario using a convenient partition of the event space. However, their optimization remains approximate. Hellström and Durisi (2021) could circumvent this problem for the particular case of sub-Gaussian losses, but the *exact* optimization of $\lambda$ for the general case remains an open question. A positive result in this direction would lift the restriction of having to optimize $\lambda$ over restricted grids or using more complex approaches.

The use of the bounded CGF assumption entails another potential drawback: both uniform control of the CGF —$\Lambda_{\boldsymbol{\theta}}(\lambda) \leq \psi(\lambda)$ for every $\boldsymbol{\theta} \in \boldsymbol{\Theta}$— and integrability assumptions as in Alquier et al. (2016) —$\mathbb{E}_\pi[\Lambda_{\boldsymbol{\theta}}(\lambda)] \leq C$ for some constant $C$—, necessarily drop information about the different concentration properties of individual models. For example, Masegosa and Ortega (2023) show that within the class of models defined by the weights of common neural networks, the behavior of their CGFs —and hence of their Cramér transforms, which control generalization error via Cramér-Chernoff bound— varies significantly. As Figure 1 shows, uniformly bounding the CGFs can result in loose *worst-case* bounds that ignore the properties of the models that interest us most.

The bounds of Seldin et al. (2012) and Haddouche et al. (2021) partially address this issue including average variances and model-dependent range functions to their bounds, which account for the fact that different models have different CGFs. However, the former only applies to bounded losses, while the latter cannot be exploited to obtain better posteriors because their model-dependent function only impacts the prior. Outside the PAC-Bayes framework, Jiao et al. (2017) generalized the bounded CGF assumption, but their result only applies to finite model sets.

Exploiting these differences among models within the same class could be an important line of research in PAC-Bayes theory. This approach would provide more informative generalization bounds, paving also the way for the design of better posteriors. This is one of the main appeals of our work.

| Metric / Model | Standard | L2 | Random | Zero |
|---|---|---|---|---|
| Train Acc. | 99.9% | 100% | 100% | 10.0% |
| Test Acc. | 84.3% | 86.6% | 10.1% | 10.0% |
| Test log-loss | 0.65 | 0.49 | 5.52 | 2.30 |
| $\ell_2$-norm | 304 | 200 | 311 | 0 |
| $\mathbb{E}_\nu\left[\|\nabla_x\ell(\boldsymbol{x},\boldsymbol{\theta})\|_2^2\right]$ | 839 | 519 | 4112 | 0 |
| $\mathbb{V}_\nu(\ell(\boldsymbol{x},\boldsymbol{\theta}))$ | 3.58 | 2.21 | 12.6 | 0 |

Figure 1: **Models with very different CGFs coexist within the same model class**. On the left, we display several metrics for InceptionV3 models trained on CIFAR10 without regularization (**Standard**) and with L2 regularization (**L2**). **Random** refers to a model learned over randomly reshuffled labels and **Zero** refers to a model where all the weights are equal to zero. For each model, the metrics include train and test accuracy, test log-loss, $\ell_2$-norm of the parameters of the model, the variance of the log-loss function, denoted $\mathbb{V}_\nu(\ell(\boldsymbol{x},\boldsymbol{\theta}))$, and the expected norm of the input-gradients, denoted $\mathbb{E}_\nu\left[\|\nabla_x\ell(\boldsymbol{x},\boldsymbol{\theta})\|_2^2\right]$. On the right, we display the estimated CGFs of each model, following Masegosa and Ortega (2023). Note how models with smaller variance $\mathbb{V}(\ell(\boldsymbol{x},\boldsymbol{\theta}))$, $\ell_2$-norm or input-gradient norm $\mathbb{E}_\nu\left[\|\nabla_x\ell(\boldsymbol{x},\boldsymbol{\theta})\|_2^2\right]$ have smaller CGFs. Bounds derived from Theorem 7 naturally exploit these differences. Experimental details in Appendix C.

**Overview and Contributions**

The main contribution of this paper is Theorem 7, a novel (oracle) PAC-Bayes bound for unbounded losses which extends the classic Cramér-Chernoff bound to the PAC-Bayesian setting. The theorem is introduced in Section 3, while Section 2 contains the necessary prerequisites. As far as we know, the proof technique based on Lemma 6 is also novel and can be of independent interest.

We discuss the applications of our main theorem in sections 4 and 5. First, we show that our bound allows *exact* optimization of the free parameter $\lambda$ incurring in a $\log n$ penalty without resorting to union-bound approaches. In the case of bounded CGFs, we recover bounds which are optimal up to that logarithmic term. We also show that versions of many well-known bounds —such as the Langford-Seeger bound— can be recovered from Theorem 7.

In Section 5, we show how our main theorem provides a general framework to deal with richer assumptions that result in novel, more informative, and potentially tighter bounds. In Theorem 11, we generalize the bounded CGF assumption so that there is a different bounding function, $\psi(\boldsymbol{\theta},\cdot)$, for the CGF of each model. We illustrate this idea in three cases: generalized sub-Gaussian losses, norm-based regularization, and input-gradient regularization based on log-Sobolev inequalities. Remarkably, the bounds in Section 5 can be minimized with respect to $\rho \in \mathcal{M}_1(\boldsymbol{\Theta})$, resulting in optimal posteriors beyond Gibbs' and opening the door to the design of novel algorithms.

Appendix A contains the proofs not included in the paper.

## 2 Preliminaries

In this section we introduce the necessary prerequisites in order to prove our main theorem. Its proof relies in controlling the concentration properties of each model in $\boldsymbol{\Theta}$ using their Cramér transform.

**Definition 4.** Let $I \subseteq \mathbb{R}$ be an interval and $f : I \to \mathbb{R}$ a convex function. The *Legendre transform* of $f$ is defined as

$$f^\star(a) := \sup_{\lambda \in I} \{\lambda a - f(\lambda)\}, \quad \forall a \in \mathbb{R}. \tag{4}$$

Following this definition, the Legendre transform of the CGF of a model $\boldsymbol{\theta} \in \boldsymbol{\Theta}$ is known as its *Cramér transform*:

$$\Lambda_{\boldsymbol{\theta}}^\star(a) := \sup_{\lambda \in [0,b)} \{\lambda a - \Lambda_{\boldsymbol{\theta}}(\lambda)\}, \quad \forall a \in \mathbb{R}. \tag{5}$$

Throughout the paper, we will use the abbreviation $gen(\boldsymbol{\theta}, D)$ to denote the *generalization gap* $L(\boldsymbol{\theta}) - \hat{L}(D, \boldsymbol{\theta})$, in order to make the notation more compact. Cramér's transform provides non-

asymptotic bounds on the right tail of $gen(\boldsymbol{\theta}, D)$ via Cramér-Chernoff's theorem —see Section 2.2 of Boucheron et al. (2013) or Section 2.2 of Dembo and Zeitouni (2009)—.

**Theorem 5** (Cramér-Chernoff). *For any $\boldsymbol{\theta} \in \boldsymbol{\Theta}$ and $a \in \mathbb{R}$,*

$$\mathbb{P}_{\nu^n}\big(gen(\boldsymbol{\theta}, D) \geq a\big) \leq e^{-n\Lambda^{\star}_{\boldsymbol{\theta}}(a)}. \tag{6}$$

*Furthermore, the inequality is asymptotically tight up to exponential factors.*

Importantly, Cramér-Chernoff's bound can be inverted to establish high-probability generalization bounds: for any $\delta \in (0, 1)$,

$$\mathbb{P}_{\nu^n}\Big(gen(\boldsymbol{\theta}, D) \leq (\Lambda^{\star}_{\boldsymbol{\theta}})^{-1}\big(\tfrac{1}{n}\log\tfrac{1}{\delta}\big)\Big) \geq 1 - \delta. \tag{7}$$

where $(\Lambda^{\star}_{\boldsymbol{\theta}})^{-1}$ is the inverse of the Cramér transform. We cannot directly use Theorem 5 to obtain PAC-Bayes bounds, because we need a bound which is uniform for every model. The following lemma will be our main technical tool for this purpose.

**Lemma 6.** *For any $\boldsymbol{\theta} \in \boldsymbol{\Theta}$ and $c \geq 0$, we have*

$$\mathbb{P}_{\nu^n}\Big(n\Lambda^{\star}_{\boldsymbol{\theta}}(gen(\boldsymbol{\theta}, D)) \geq c\Big) \leq \mathbb{P}_{X \sim \exp(1)}\Big(X \geq c\Big).$$

This way of controlling the survival function of $\Lambda^{\star}_{\boldsymbol{\theta}}(gen(\boldsymbol{\theta}, D))$ for every $\boldsymbol{\theta} \in \boldsymbol{\Theta}$ will allow us to bound an exponential moment term in our main theorem.

## 3 PAC-Bayes-Chernoff bound

As we hinted above, instead of directly introducing boundedness conditions on the loss or its CGF, we stay in the realm of oracle bounds, aiming to provide a flexible starting point for diverse applications. A key element of our theoretical approach is the averaging of the CGFs with respect to a posterior distribution. For any posterior distribution $\rho \in \mathcal{M}_1(\boldsymbol{\Theta})$, we may consider the expectation of the CGF, $\mathbb{E}_\rho[\Lambda_{\boldsymbol{\theta}}(\lambda)]$, as in Jiao et al. (2017). In analogy to the standard definition, we define the *Cramér transform of a posterior distribution $\rho$* as the following function:

$$\Lambda^{\star}_\rho(a) := \sup_{\lambda \in [0,b)} \{\lambda a - \mathbb{E}_\rho[\Lambda_{\boldsymbol{\theta}}(\lambda)]\}, \quad a \in \mathbb{R}. \tag{8}$$

Since the CGFs $\Lambda_{\boldsymbol{\theta}}(\lambda)$ are convex and continuously differentiable with respect to $\lambda$, their expectation $\mathbb{E}_\rho[\Lambda_{\boldsymbol{\theta}}(\lambda)]$ retains the same properties. Hence according to Lemma 2.4 in Boucheron et al. (2013), the (generalized) inverse of $\Lambda^{\star}_\rho$ exists and can be written as

$$\big(\Lambda^{\star}_\rho\big)^{-1}(s) = \inf_{\lambda \in [0,b)}\left\{\frac{s + \mathbb{E}_\rho[\Lambda_{\boldsymbol{\theta}}(\lambda)]}{\lambda}\right\}. \tag{9}$$

With these definitions in hand, we are ready to introduce our main result, a novel (oracle) PAC-Bayes bound for unbounded losses:

**Theorem 7** (PAC-Bayes-Chernoff bound). *Let $\pi \in \mathcal{M}_1(\boldsymbol{\Theta})$ be any prior independent of $D$. Then, for any $\delta \in (0, 1)$, with probability at least $1 - \delta$ over draws of $D \sim \nu^n$,*

$$\mathbb{E}_\rho[L(\boldsymbol{\theta})] \leq \mathbb{E}_\rho[\hat{L}(D, \boldsymbol{\theta})] + \inf_{\lambda \in [0,b)}\left\{\frac{KL(\rho|\pi) + \log\frac{n}{\delta}}{\lambda(n-1)} + \frac{\mathbb{E}_\rho[\Lambda_{\boldsymbol{\theta}}(\lambda)]}{\lambda}\right\}$$

*simultaneously for every $\rho \in \mathcal{M}_1(\boldsymbol{\Theta})$.*

The above result gives an oracle PAC-Bayes analogue to the Cramér-Chernoff's bound of equation (7), because the second term of the right hand side of the inequality corresponds with the inverse of $\Lambda^{\star}_\rho$, as described in equation (9). Theorem 7 shows that bounds like the one given in Theorem 3 can hold simultaneously for every $\lambda > 0$ —and hence optimized with respect to $\lambda$— by paying a $\log(n)$ penalty, virtually the same as if we were optimizing over a uniform grid of size $n$ using union-bound arguments. Although the dependence in union bound arguments can be improved to $\log(\log(n))$ (Alquier, 2024), our optimization is exact.

Furthermore, Theorem 7 also shows that the posterior minimizing its right-hand-side is involved in a three-way trade-off: firstly, $\rho$ must explain the training data due to $\mathbb{E}_\rho[\hat{L}(D, \boldsymbol{\theta})]$; secondly, it must be close to the prior due to the KL term $KL(\rho|\pi)$; and lastly, it must place its density in models with a lower CGF due to the $\mathbb{E}_\rho[\Lambda_{\boldsymbol{\theta}}(\lambda)]$ term. We note that the first two elements are standard on most PAC-Bayesian bounds, but the role of the CGF in defining an optimal posterior $\rho$ is novel compared to previous bounds. We explore the implications of this fact in Section 5.

## 4  Relation with previous bounds

We first relate Theorem 7 to previous bounds. As a first application, we show how some well-known PAC-Bayes bounds can be recovered from ours. When the distribution of the loss is known, we can often compute its Cramér transform. This happens to be the case with the zero-one loss, where we recover Langford-Seeger's bound (Seeger, 2002, Theorem 1).

**Corollary 8.** *Let $\ell$ be the $0 - 1$ loss and $\pi \in \mathcal{M}_1(\boldsymbol{\Theta})$ be any prior independent of $D$. Then, for any $\delta \in (0, 1)$, with probability at least $1 - \delta$ over draws of $D \sim \nu^n$,*

$$kl\left(\mathbb{E}_\rho[\hat{L}(\boldsymbol{\theta}, D)], \mathbb{E}_\rho[L(\boldsymbol{\theta})]\right) \leq \frac{KL(\rho|\pi) + \log \frac{n}{\delta}}{n - 1},$$

*simultaneously for every $\rho \in \mathcal{M}_1(\boldsymbol{\Theta})$.*

The dependence on $n$ in Corollary 8 was further improved by Maurer (2004). However, our version is enough to illustrate the role played by Cramér transforms in obtaining tight PAC-Bayes bounds, which is a recent line of work by Foong et al. (2021) and Hellström and Guedj (2024).

When the loss is of bounded CGF —recall Definition 2—, Theorem 7 results in a generalization of Theorem 3, where the bound holds simultaneously for every $\lambda \in (0, b)$ with a $\log n$ penalty.

**Corollary 9.** *Let $\ell$ be a loss function with $\psi$-bounded CGF. Let $\pi \in \mathcal{M}_1(\boldsymbol{\Theta})$ be any prior independent of $D$. Then, for any $\delta \in (0, 1)$, with probability at least $1 - \delta$ over draws of $D \sim \nu^n$,*

$$\mathbb{E}_\rho[L(\boldsymbol{\theta})] \leq \mathbb{E}_\rho[\hat{L}(D, \boldsymbol{\theta})] + \inf_{\lambda \in [0, b)}\left\{\frac{KL(\rho|\pi) + \log \frac{n}{\delta}}{\lambda(n - 1)} + \frac{\psi(\lambda)}{\lambda}\right\},$$

*simultaneously for every $\rho \in \mathcal{M}_1(\boldsymbol{\Theta})$.*

*Proof.* Directly follows from the definition of $\psi$-bounded CGF and Theorem 7. □

Corollary 9 is the first PAC-Bayes bound that allows exact optimization of the free parameter $\lambda \geq 0$ for the general case of losses with bounded CGF without resorting to union-bound approaches (Seldin et al., 2012; Rodríguez-Gálvez et al., 2024), which cannot guarantee an exact minimization. Although in the particular case of sub-Gaussian losses the $\log n$ penalty is worse than that in Hellström and Durisi (2021), when $KL(\rho|\pi) \geq \frac{\sqrt{6n/e}}{\pi} - 1$, Corollary 9 is tighter than the general Theorem 14 in Rodríguez-Gálvez et al. (2024). We instantiate Corollary 9 in the case of sub-Gaussian and sub-gamma losses in the Appendix B.

## 5  PAC-Bayes bounds under model-dependent assumptions

As discussed in the introduction, most boundedness conditions used to bound the exponential moment term in Theorem 7 discard information about the statistical properties of individual models. In this section, we show that the structure of Theorem 7 allows for a more fine-grained control of the CGFs, resulting in potentially tighter and more informative bounds. We start by generalizing Definition 2.

**Definition 10** (Model-dependent bounded CGF). A loss function $\ell$ has model-dependent bounded CGF if for each $\boldsymbol{\theta} \in \boldsymbol{\Theta}$, there is a convex and continuously differentiable function $\psi(\boldsymbol{\theta}, \lambda)$ such that $\psi(\boldsymbol{\theta}, 0) = \psi'(\boldsymbol{\theta}, 0) = 0$ and for any $\lambda \in [0, b)$,

$$\Lambda_{\boldsymbol{\theta}}(\lambda) := \log \mathbb{E}_\nu\left[e^{\lambda(L(\boldsymbol{\theta}) - \ell(\boldsymbol{x}, \boldsymbol{\theta}))}\right] \leq \psi(\boldsymbol{\theta}, \lambda). \tag{10}$$

As motivated in Figure 1 and in opposition to Definition 2, this new condition acknowledges the possibility of having different bounding functions, $\psi(\boldsymbol{\theta}, \lambda)$, for each $\Lambda_{\boldsymbol{\theta}}(\lambda)$.

Using this definition, we can easily exploit Theorem 7 to derive the following bound:

**Theorem 11.** *Let $\ell$ be a loss function satisfying Definition 10. Let $\pi \in \mathcal{M}_1(\boldsymbol{\Theta})$ be any prior independent of $D$. Then, for any $\delta \in (0,1)$, with probability at least $1 - \delta$ over draws of $D \sim \nu^n$,*

$$\mathbb{E}_\rho[L(\boldsymbol{\theta})] \leq \mathbb{E}_\rho[\hat{L}(D, \boldsymbol{\theta})] + \inf_{\lambda \in [0,b)} \left\{ \frac{KL(\rho|\pi) + \log \frac{n}{\delta}}{\lambda(n-1)} + \frac{\mathbb{E}_\rho[\psi(\boldsymbol{\theta}, \lambda)]}{\lambda} \right\}, \tag{11}$$

*simultaneously for every $\rho \in \mathcal{M}_1(\boldsymbol{\Theta})$.*

*Proof.* The proof is analogue to that of Corollary 9. $\qquad\square$

This theorem can be understood as a PAC-Bayesian version of Theorem 2 in Jiao et al. (2017), where we allow infinite model classes. Note that if we tried to exploit this model-dependent CGF assumption on other oracle bounds, as the one shown in Theorem 1, we would end with an empirical bound where the $\psi(\boldsymbol{\theta}, \lambda)$ term would be *exponentially averaged* by the prior, $\ln \mathbb{E}_\pi[e^{\psi(\boldsymbol{\theta}, \lambda)}]$, instead of having the more amenable term $\mathbb{E}_\rho[\psi(\boldsymbol{\theta}, \lambda)]$, which directly impacts on the choice of the optimal posterior $\rho^\star$.

As discussed in Section 3, the posterior distribution in Theorem 11 is involved in a three-way trade-off which has the potential to result in tighter bounds and the design of better posteriors. It is worth noting that the posterior minimizing the bound in Theorem 11 is not the standard Gibbs posterior.

**Proposition 12.** *If we fix some $\lambda > 0$, the bound in Theorem 11 can be minimized with respect to $\rho \in \mathcal{M}_1(\boldsymbol{\Theta})$. The optimal posterior is*

$$\rho^*(\boldsymbol{\theta}) \propto \pi(\boldsymbol{\theta}) \exp \left\{ -(n-1)\lambda \hat{L}(D, \boldsymbol{\theta}) - (n-1)\psi(\boldsymbol{\theta}, \lambda) \right\}. \tag{12}$$

Observe that under the posterior in Proposition 12, the *maximum a posteriori* (MAP) estimate is

$$\boldsymbol{\theta}_{\text{MAP}} = \arg\min_{\boldsymbol{\theta} \in \boldsymbol{\Theta}} \left\{ \hat{L}(D, \boldsymbol{\theta}) + \frac{1}{\lambda} \psi(\boldsymbol{\theta}, \lambda) - \frac{1}{\lambda(n-1)} \ln \pi(\boldsymbol{\theta}) \right\}.$$

As we will illustrate below, the extra term, $\psi(\boldsymbol{\theta}, \lambda)$, can often be understood as a regularizer. In what remains, we exemplify the general recipe of Theorem 11 in several cases. To start providing concrete intuition, we instantiate Corollary 11 in the case of sub-Gaussian losses.

### Generalized sub-Gaussian losses

It is well known that if $X$ is a $\sigma^2$-sub-Gaussian random variable, we have $\mathbb{V}(X) \leq \sigma^2$ (Arbel et al., 2020). In many cases, it might not be reasonable to bound $\mathbb{V}_\nu(\ell(\boldsymbol{\theta}, X)) \leq \sigma^2$ for every $\boldsymbol{\theta} \in \boldsymbol{\Theta}$, because the variance of the loss function highly depends on the particular model, as illustrated in Figure 1. This is where Corollary 11 comes into play: we may assume that $\ell(\boldsymbol{\theta}, X)$ is $\sigma(\boldsymbol{\theta})^2$-sub-Gaussian for each $\boldsymbol{\theta} \in \boldsymbol{\Theta}$. In this case the variance proxy $\sigma(\boldsymbol{\theta})^2$ is specific for each model, leading to the following bound:

**Corollary 13.** *Assume the loss $\ell(\boldsymbol{\theta}, X)$ is $\sigma^2(\boldsymbol{\theta})$-sub-Gaussian. Let $\pi \in \mathcal{M}_1(\boldsymbol{\Theta})$ be any prior independent of $D$. Then, for any $\delta \in (0,1)$, with probability at least $1 - \delta$ over draws of $D \sim \nu^n$,*

$$\mathbb{E}_\rho[L(\boldsymbol{\theta})] \leq \mathbb{E}_\rho[\hat{L}(D, \boldsymbol{\theta})] + \sqrt{2\mathbb{E}_\rho[\sigma(\boldsymbol{\theta})^2] \frac{KL(\rho|\pi) + \log \frac{n}{\delta}}{n-1}}, \tag{13}$$

*simultaneously for every $\rho \in \mathcal{M}_1(\boldsymbol{\Theta})$.*

*Proof.* Use Theorem 11 and the fact that $\psi(\boldsymbol{\theta}, \lambda) = \frac{\lambda^2 \sigma^2(\boldsymbol{\theta})}{2}$. Then optimize $\lambda$. $\qquad\square$

This result generalizes sub-Gaussian PAC-Bayes bounds —Corollary 2 in Hellström and Durisi (2021) or Corollary 19— and shows that posteriors favoring models with small variance-proxy, $\sigma^2(\boldsymbol{\theta})$, generalize better. It is, therefore, potentially tighter than previous results, because the $\sigma^2$

factor in standard sub-Gaussian bounds is a *worst-case constant*, while Corollary 13 exploits the fact that some models have much smaller variance-proxy than others.

Analogous bounds can be straightforwardly derived for generalized sub-gamma or sub-exponential losses, but Theorem 11 is not limited to explicit tail assumptions on the loss. The following subsections explore model-dependent assumptions on $\Lambda_{\boldsymbol{\theta}}(\lambda)$ that result in novel PAC-Bayes bounds involving well-known regularization techniques.

**L2 regularization**

We now introduce bounds based on the norm of the model parameters. For that purpose we use the following standard assumption in machine learning (Li and Orabona, 2019).

*Assumption* 1 (Parameter Lipschitz). The loss function $\ell(\boldsymbol{x}, \boldsymbol{\theta})$ is $M$-Lipschitz with respect to $\boldsymbol{\theta}$, that is, for any $\boldsymbol{\theta} \in \boldsymbol{\Theta}$ and any $\boldsymbol{x} \in \mathcal{X}$ we have $\|\nabla_{\boldsymbol{\theta}} \ell(\boldsymbol{x}, \boldsymbol{\theta})\|_2^2 \leq M$.

If the model class is parametrized in such a way that the model with null parameter vector has null variance —i.e., $\mathbb{V}_{\nu}(\ell(\boldsymbol{x}, 0)) = 0$, which is the case of a neural net with null weights—, then, as shown in Masegosa and Ortega (2023), we can derive the following model-dependent bound for the CGF:

$$\Lambda_{\boldsymbol{\theta}}(\lambda) \leq 2M\lambda^2 \|\boldsymbol{\theta}\|_2^2. \tag{14}$$

This case further illustrates the idea we motivated in the introduction: bounding the CGFs with model-dependent proxies for generalization. Figure 1 illustrates how models with smaller norm has increasingly smaller CGFs.

Using Equation (14) we obtain the following generalization bound penalizing model's L2-norm:

**Corollary 14.** *If Assumption 1 holds, then for any prior distribution $\pi \in \mathcal{M}_1(\boldsymbol{\Theta})$ independent of $D$ and any $\delta \in (0, 1)$, with probability at least $1 - \delta$ over draws $D \sim \nu^n$,*

$$\mathbb{E}_{\rho}[L(\boldsymbol{\theta})] \leq \mathbb{E}_{\rho}[\hat{L}(D, \boldsymbol{\theta})] + \sqrt{2M\mathbb{E}_{\rho}\left[\|\boldsymbol{\theta}\|_2^2\right] \frac{KL(\rho|\pi) + \log \frac{n}{\delta}}{n - 1}}, \tag{15}$$

*simultaneously for every $\rho \in \mathcal{M}_1(\boldsymbol{\Theta})$.*

*Proof.* The result follows from using $\psi(\boldsymbol{\theta}, \lambda) = 2M\lambda^2 \|\boldsymbol{\theta}\|_2^2$ in Theorem 11 and optimizing $\lambda$. $\square$

Corollary 14 shows that models with smaller parameter norms generalize better, and provides PAC-Bayesian certificates for norm-based regularization. Many other PAC-Bayesian bounds contains different kind of parameter norms (Germain et al., 2009, 2016; Neyshabur et al., 2017), but their parameter norm term is always introduced through the prior —e.g., using a zero-centered Gaussian prior distribution—. The novelty here is that this parameter norm term appears independently of the prior as a result of Theorem 11.

According to Proposition 12, the MAP estimate of the optimal posterior is, $\boldsymbol{\theta}_{\text{MAP}} = \arg\min_{\boldsymbol{\theta}} \left\{ \hat{L}(D, \boldsymbol{\theta}) + \frac{2M}{\lambda} \|\boldsymbol{\theta}\|_2^2 - \frac{1}{\lambda(n-1)} \ln \pi(\boldsymbol{\theta}) \right\}$, which is the result of L2-regularization with tradeoff parameter $\frac{2M}{\lambda}$.

**Gradient-based regularization**

We finish this section providing novel bounds based on log-Sobolev inequalities that include a gradient term penalizing the sensitivity of models to small changes in the input data. First, let us simplify the notation for this section. We use $\|\nabla_{\boldsymbol{x}} \ell\|_2^2 := \mathbb{E}_{\nu} \|\nabla_{\boldsymbol{x}} \ell(\boldsymbol{x}, \boldsymbol{\theta})\|_2^2$ and $\|\widehat{\nabla}_{\boldsymbol{x}} \ell\|_2^2 := \frac{1}{n} \sum_{i=1}^n \|\nabla_{\boldsymbol{x}} \ell(\boldsymbol{x}_i, \boldsymbol{\theta})\|_2^2$ to denote the expected and empirical (squared) gradient norms of the loss.

Including a gradient-based penalization is the idea behind input-gradient regularization (Varga et al., 2017), which minimizes an objective function of the form $\hat{L}(D, \boldsymbol{\theta}) + \frac{1}{k} \|\widehat{\nabla}_{\boldsymbol{x}} \ell\|_2^2$, where $k > 0$ is a trade-off parameter. This approach is often used to make models more robust against disturbances in input data and adversarial attacks (Ross and Doshi-Velez, 2018; Finlay and Oberman, 2021).

We make the connection between PAC-Bayes bounds and gradient norms assuming that $\nu$ and $\ell$ satisfy certain log-Sobolev inequality (Chafaï, 2004).

*Assumption* 2 (log-Sobolev). For any $\boldsymbol{\theta} \in \boldsymbol{\Theta}$ and any $\lambda > 0$, we have

$$\Lambda_{\boldsymbol{\theta}}(\lambda) \leq \frac{C}{2}\lambda^2 \|\nabla_{\boldsymbol{x}}\ell\|_2^2, \text{ for some } C > 0.$$

For example, Assumption 2 holds when $\nu$ is strictly uniformly log-concave, as shown in Corollary 2.1 of Chafaï (2004) —this case includes the Gaussian and Weibull densities (Saumard and Wellner, 2014)—. We also try to empirically verify Assumption 2 for certain class of neural networks in Appendix C.2. Assumption 2 is going to be our model-dependent bound on the CGF. We first provide a bound for expected gradients.

**Theorem 15.** *If Assumption 2 is satisfied, then for any prior distribution $\pi \in \mathcal{M}_1(\boldsymbol{\Theta})$ independent of $D$ and any $\delta_1 \in (0,1)$, with probability at least $1 - \delta_1$ over draws $D \sim \nu^n$,*

$$\mathbb{E}_{\rho}[L(\boldsymbol{\theta})] \leq \mathbb{E}_{\rho}[\hat{L}(D,\boldsymbol{\theta})] + \sqrt{2C\mathbb{E}_{\rho}\left[\|\nabla_{\boldsymbol{x}}\ell\|_2^2\right]\frac{KL(\rho\|\pi) + \log\frac{n}{\delta_1}}{n-1}} \tag{16}$$

*simultaneously for every $\rho \in \mathcal{M}_1(\boldsymbol{\Theta})$.*

*Proof.* Follows from the application Assumption 2 to Theorem 11 and the optimization of $\lambda$. $\square$

This bound shows that —under certain regularity conditions— posteriors favoring models with smaller expected gradients, $\|\nabla_{\boldsymbol{x}}\ell\|_2^2$, generalize better, which is the heuristic behind input-gradient regularization (Varga et al., 2017). However, Theorem 15 is still an oracle bound. We can obtain a new, fully empirical one if we concatenate Corollary 15 with a PAC-Bayes concentration bound for $\mathbb{E}_{\rho}\left[\|\nabla_{\boldsymbol{x}}\ell\|_2^2\right]$. This can be done if we assume that the loss is Lipschitz w.r.t. the input-data.

*Assumption* 3 (Input-data Lipschitz). For any $\boldsymbol{\theta} \in \boldsymbol{\Theta}$ and for any $\boldsymbol{x} \in \mathcal{X}$, we have $\|\nabla_{\boldsymbol{x}}\ell(\boldsymbol{x},\boldsymbol{\theta})\|_2^2 \leq L$.

Assumption 3 is satisfied in standard deep neural network architectures, and the Lipschitz constant can be efficiently estimated (Virmaux and Scaman, 2018; Fazlyab et al., 2019).

**Theorem 16.** *If Assumptions 2 and 3 are satisfied, then for any prior distribution $\pi \in \mathcal{M}_1(\boldsymbol{\Theta})$ independent of $D$ and any $\delta \in (0,1)$; with probability at least $1-\delta$ over draws of $D \sim \nu^n$,*

$$\mathbb{E}_{\rho}[L(\boldsymbol{\theta})] \leq \mathbb{E}_{\rho}[\hat{L}(D,\boldsymbol{\theta})] + \sqrt{2C\mathbb{E}_{\rho}\left[\|\widehat{\nabla}_{\boldsymbol{x}}\ell\|_2^2\right] \cdot K(\rho,\pi,n,\delta) + \sqrt{2}CL \cdot K(\rho,\pi,n,\delta)^{\frac{3}{2}}}$$

*simultaneously for every $\rho \in \mathcal{M}_1(\boldsymbol{\Theta})$, where $K(\rho,\pi,n,\delta) := \frac{KL(\rho|\pi) + \log\frac{2n}{\delta}}{n-1}$.*

With minimal modifications, the result above also holds under the assumption that on-average gradients are bounded, $\|\nabla_{\boldsymbol{x}}\ell\|_2^2 \leq g$, or when $\|\nabla_{\boldsymbol{x}}\ell(\boldsymbol{\theta}, X)\|_2^2$ are sub-Gaussian.

Similar bounds with input-gradients were first introduced by Gat et al. (2022) under the assumption that the underlying distribution of data was a mixture of Gaussians plus a technical "per-label loss balance" assumption. However, Theorem 16 is, as far as we know, the first empirical PAC-Bayes bound for input-gradients. Furthermore, in contrast with Gat et al. (2022), our bounds optimize the free parameter $\lambda$ and explicitly relate the gradients with the generalization ability of the posterior $\rho \in \mathcal{M}_1(\boldsymbol{\Theta})$. The recent work of Haddouche et al. (2024) also includes gradient terms in their bounds, but they are gradients with respect to the model parameter, not input-gradients.

As for the optimal posterior $\rho^*$, if we minimize the bound in Theorem 15 for a fixed $\lambda > 0$, the MAP estimate in Proposition 12 is $\boldsymbol{\theta}_{\text{MAP}} = \arg\min_{\boldsymbol{\theta}}\{\hat{L}(D,\boldsymbol{\theta}) + \lambda\frac{C}{2}\|\nabla_{\boldsymbol{x}}\ell\|_2^2 - \frac{1}{\lambda(n-1)}\ln\pi(\boldsymbol{\theta})\}$, which is the result of input-gradient regularization with trade-off parameter $\frac{\lambda C}{2}$.

In conclusion, the approach suggested by Theorem 11 not only provides novel insights —in the form of tight bounds or PAC-Bayesian interpretations of previously known algorithms—, it also hints towards the design of new regularized learning algorithms with solid theoretical guarantees.

## 6 Conclusion

We derived a novel PAC-Bayes oracle bound using basic properties of the Cramér transform —Theorem 7—. In general, our work aligns with very recent literature (Rodríguez-Gálvez et al.,

2024; Hellström et al., 2023) that highlights the importance of Cramér transforms in the quest for tight PAC-Bayes bounds. This bound has the potential to be a stepping stone in the development of novel, tighter empirical PAC-Bayes bounds for unbounded losses. Firstly, because it allows exact optimization of the free parameter $\lambda > 0$ without the need for more convoluted union-bound approaches. But, more relevantly, because it allows the introduction of flexible, fine-grained, model-dependent assumptions for bounding the CGF —Theorem 11— which results in optimal distributions beyond Gibbs' posterior. The importance and wide applicability of this result have been illustrated with three model-dependent assumptions: generalized sub-Gaussian losses, bounds based on parameter norms, and input-gradients based on log-Sobolev inequalities. In the last case we introduce PAC-Bayes bounds including empirical input-gradients norms.

**Limitations and future work**

A limitation of our approach is that the we are implicitly assuming that $\ell(\boldsymbol{\theta}, X)$ is light-tailed (equivalently, sub-exponential), as in every Cramér-Chernoff bound. This is only partially true. Although there are specific studies for heavy-tailed losses (Alquier and Guedj, 2018; Holland, 2019; Haddouche and Guedj, 2023; Chugg et al., 2023), we avoid this limitation because we are only interested in the right tail of $gen(\boldsymbol{\theta}, D)$ —that is, $\mathbb{P}_\nu(L(\boldsymbol{\theta}) - \hat{L}(D, \boldsymbol{\theta}) \geq a)$ for $a \geq 0$—. This is done by defining the CGF only for $\lambda \geq 0$. In this way, as we show in Remark 17, the finiteness of $\mathbb{E}_\rho[L(\boldsymbol{\theta})]$ guarantees the existence of $\mathbb{E}_\rho[\Lambda_{\boldsymbol{\theta}}(\lambda)]$ in Theorem 7. This approach is motivated by the fact that most state-of-the-art models lie in the interpolating regime, hence $\mathbb{P}_\nu(\hat{L}(D, \boldsymbol{\theta}) - L(\boldsymbol{\theta}) \geq a)$ for $a \geq 0$ has less practical importance. However, in the case where one is looking for two-tailed bounds, our work is restricted to sub-exponential losses. See the discussion on Section 4.2.1 of Zhang et al. (2024).

Given the generality of Theorem 7, the dependence of the logarithmic penalty term in the bound may be suboptimal in certain cases, as discussed in Section 4. Going beyond Lemma 6 and improving this dependence is one of the main open tasks.

Our results provide a systematic approach for integrating general model-dependent CGF bounds into PAC-Bayes theory, and can open exciting new research lines: they can provide a PAC-Bayesian extension of the work of Masegosa and Ortega (2023), which studies the effects of different regularization techniques and the role of invariant architectures, data augmentation and over-parametrization in generalization. Since our bounds can be applied to unbounded losses such as the log-loss, it could be interesting to apply them in a Bayesian setting (Germain et al., 2016) in order to study the relation between marginal likelihood and generalization (Lotfi et al., 2022).

## Acknowledgments and disclosure of funding

IC acknowledges financial support from the Basque Government through the IKUR program. LO acknowledges financial support from project PID2022-139856NB-I00 funded by MCIN/ AEI/ 10.13039/501100011033/FEDER, UE and from the Autonomous Community of Madrid (ELLIS Unit Madrid). AP acknowledges financial support by the Basque Government through the BERC 2022-2025 program and Elkartek program (KK-2023/00038), and by the Ministry of Science and Innovation: BCAM Severo Ochoa accreditation CEX2021-001142-S/MICIN/AEI/10.13039/501100011033. AM acknowledges financial support from Grant PID2022-139293NB-C31 funded by MCIN/AEI/10.13039/501100011033 and by ERDF A way of making Europe.

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

# A  Proofs

We first gather some standard properties of CGFs and Cramér transforms.

Let $\mathcal{D}_{\Lambda_{\boldsymbol{\theta}}} := \{\lambda \in \mathbb{R}_+ \mid \Lambda_{\boldsymbol{\theta}}(\lambda) < \infty\}$. Under the assumption that there is some $\lambda > 0$ such that $\Lambda_{\boldsymbol{\theta}}(\lambda) < \infty$, we have $\mathcal{D}_{\Lambda_{\boldsymbol{\theta}}} = [0, b)$ for some $b > 0$. Furthermore, $\Lambda_{\boldsymbol{\theta}}$ is strictly convex and $\mathcal{C}^{\infty}$ on $(0, b)$ (Boucheron et al., 2013, Section 2.2).

In particular, since for any $\lambda > 0$ we have $\Lambda_{\boldsymbol{\theta}}(\lambda) := \log \mathbb{E}_{\nu} \left[ e^{\lambda(L(\boldsymbol{\theta}) - \ell(x,\boldsymbol{\theta}))} \right] \leq \log \mathbb{E}_{\nu} \left[ e^{\lambda L(\boldsymbol{\theta})} \right] = \lambda L(\boldsymbol{\theta}) < \infty$ because $\ell \geq 0$, we verify that $\mathcal{D}_{\Lambda_{\boldsymbol{\theta}}} = [0, \infty)$ and that $\Lambda_{\boldsymbol{\theta}}$ is strictly convex and $\mathcal{C}^{\infty}$ on $(0, \infty)$.

*Remark* 17. Observe that the previous proof also shows that $\mathbb{E}_{\rho}[L(\boldsymbol{\theta})] < \infty$ is enough to guarantee that $\mathbb{E}_{\rho}[\Lambda_{\boldsymbol{\theta}}(\lambda)]$ is finite for every $\lambda \geq 0$.

The following technical result will be important for the proof of Lemma 6. It combines the previous remarks with Lemma 2.2.5(c) of Dembo and Zeitouni (2009).

**Lemma 18.** $\Lambda_{\boldsymbol{\theta}}^{\star}(a) < \infty$ *for any* $a \in [0, L(\boldsymbol{\theta}))$*. Furthermore,* $\Lambda_{\boldsymbol{\theta}}^{\star}$ *is differentiable in* $(0, L(\boldsymbol{\theta}))$*.*

*Proof.* We start by proving the finiteness condition. Since $\Lambda_{\boldsymbol{\theta}}$ is strictly convex on $(0, \infty)$, $\Lambda_{\boldsymbol{\theta}}'$ is also strictly increasing on $(0, \infty)$. This means that $\Lambda_{\boldsymbol{\theta}}'$ is a bijection from $(0, \infty)$ to

$$\left( \inf_{\lambda \in (0,\infty)} \Lambda_{\boldsymbol{\theta}}'(\lambda), \sup_{\lambda \in (0,\infty)} \Lambda_{\boldsymbol{\theta}}'(\lambda) \right). \text{ In other words, for every } a \in \left( \inf_{\lambda \in (0,\infty)} \Lambda_{\boldsymbol{\theta}}'(\lambda), \sup_{\lambda \in (0,\infty)} \Lambda_{\boldsymbol{\theta}}'(\lambda) \right),$$

the equation $a = \Lambda_{\boldsymbol{\theta}}'(\lambda)$ has a unique solution $\lambda_a \in (0, \infty)$.

Now consider the function $g(\lambda) = \lambda a - \Lambda_{\boldsymbol{\theta}}(\lambda)$. The previous observation means that $g'(\lambda_a) = 0$, and since $g$ is concave, $\Lambda_{\boldsymbol{\theta}}^{\star}(a) = \sup_{\lambda \in (0,\infty)} \{\lambda a - \Lambda_{\boldsymbol{\theta}}(\lambda)\} = g(\lambda_a) < \infty$.

It only remains to prove that $\inf_{\lambda \in (0,\infty)} \Lambda_{\boldsymbol{\theta}}'(\lambda) = 0$ and $\sup_{\lambda \in (0,\infty)} \Lambda_{\boldsymbol{\theta}}'(\lambda) = L(\boldsymbol{\theta})$. The first part follows from the fact that $\Lambda_{\boldsymbol{\theta}}'(0) = 0$. The second one is equivalent to proving

$$\lim_{\lambda \to \infty} \Lambda_{\boldsymbol{\theta}}'(\lambda) = \mathrm{ess\,sup}(L(\boldsymbol{\theta}) - \ell(X, \boldsymbol{\theta})),$$

where, without loss of generality, we assume that $\mathrm{ess\,inf}\,\ell(X, \boldsymbol{\theta}) = 0$. This is also a standard property of CGFs (see, for example, this proof).

Finally, differentiability of $\Lambda_{\boldsymbol{\theta}}^{\star}$ follows from its finiteness in $[0, L(\boldsymbol{\theta}))$ and Lemma 6(5) in Gantert et al. (2016). □

With the tools we gathered above, we are ready to prove our main technical lemma:

**Lemma 6.** *For any* $\boldsymbol{\theta} \in \boldsymbol{\Theta}$ *and* $c \geq 0$*, we have*

$$\mathbb{P}_{\nu^n} \left( n \Lambda_{\boldsymbol{\theta}}^{\star}(gen(\boldsymbol{\theta}, D)) \geq c \right) \leq \mathbb{P}_{X \sim \exp(1)} \left( X \geq c \right).$$

*Proof.* The proof relies in the properties of generalized inverses. Consider

$$\mathbb{P}_{D \sim \nu^n} \left( n \Lambda_{\boldsymbol{\theta}}^{\star}(gen(\boldsymbol{\theta}, D)) \geq a \right)$$

for $a \geq 0$. We will separately consider three cases:

Case 1: $a/n < \sup_{x \in (0, L(\boldsymbol{\theta}))} \Lambda_{\boldsymbol{\theta}}^{\star}(x)$.

By Proposition 1(5) in Embrechts and Hofert (2013), we have

$$\mathbb{P}_{D \sim \nu^n} \left( n \Lambda_{\boldsymbol{\theta}}^{\star}(gen(\boldsymbol{\theta}, D)) \geq a \right) \leq \mathbb{P}_{D \sim \nu^n} \left( gen(\boldsymbol{\theta}, D) \geq (\Lambda_{\boldsymbol{\theta}}^{*})^{-1}(a/n) \right),$$

and using the Cramér-Chernoff bound on the right-hand side we obtain

$$\mathbb{P}_{D \sim \nu^n} \left( n \Lambda_{\boldsymbol{\theta}}^{\star}(gen(\boldsymbol{\theta}, D)) \geq a \right) \leq e^{-n \Lambda_{\boldsymbol{\theta}}^{\star} \left( (\Lambda_{\boldsymbol{\theta}}^{*})^{-1}(a/n) \right)}.$$

This results in

$$\mathbb{P}_{D\sim\nu^n}\left(n\Lambda_{\boldsymbol{\theta}}^{\star}(gen(\boldsymbol{\theta},D))\geq a\right)\leq e^{-a}$$

because $\Lambda_{\boldsymbol{\theta}}^{\star}$ is continuous (in fact differentiable, see Lemma 18) on $a/n$ and we can apply Proposition 1(4) in Embrechts and Hofert (2013).

Case 2: $a/n = \sup_{x\in(0,L(\boldsymbol{\theta}))}\Lambda_{\boldsymbol{\theta}}^{\star}(x)$.

$$\begin{aligned}
\mathbb{P}_{D\sim\nu^n}\left(n\Lambda_{\boldsymbol{\theta}}^{\star}(gen(\boldsymbol{\theta},D))\geq a\right) &= \mathbb{P}_{D\sim\nu^n}\left(gen(\boldsymbol{\theta},D)=L(\boldsymbol{\theta})\right)\\
&= \mathbb{P}_{X\sim\nu}\left(\ell(X,\boldsymbol{\theta})=0\right)^n\\
&= e^{-n\Lambda_{\boldsymbol{\theta}}^{\star}(L(\boldsymbol{\theta}))}.
\end{aligned}$$

If $\Lambda_{\boldsymbol{\theta}}^{\star}(L(\boldsymbol{\theta})) < \infty$, then $a/n = \Lambda_{\boldsymbol{\theta}}^{\star}(L(\boldsymbol{\theta}))$ by the lower semi-continuity of $\Lambda_{\boldsymbol{\theta}}^{\star}$, and the result holds. Otherwise it is trivially true.

Case 3: $a/n > \sup_{x\in(0,L(\boldsymbol{\theta}))}\Lambda_{\boldsymbol{\theta}}^{\star}(x)$. In this case

$$\mathbb{P}_{D\sim\nu^n}\left(n\Lambda_{\boldsymbol{\theta}}^{\star}(gen(\boldsymbol{\theta},D))\geq a\right) = \mathbb{P}_{D\sim\nu^n}\left(\Lambda_{\boldsymbol{\theta}}^{\star}(gen(\boldsymbol{\theta},D))=\infty\right).$$

But $\Lambda_{\boldsymbol{\theta}}^{\star}(gen(\boldsymbol{\theta},D)) = \infty$ can only happen if $gen(\boldsymbol{\theta},D) = L(\boldsymbol{\theta})$ and $\Lambda_{\boldsymbol{\theta}}^{\star}(L(\boldsymbol{\theta})) = \infty$, and this means that $\mathbb{P}_{D\sim\nu^n}\left(gen(\boldsymbol{\theta},D)=L(\boldsymbol{\theta})\right) = 0$.

This concludes the proof. $\qquad\square$

Now we list every result whose proof is not included in the paper.

**Theorem 7** (PAC-Bayes-Chernoff bound). *Let $\pi \in \mathcal{M}_1(\boldsymbol{\Theta})$ be any prior independent of $D$. Then, for any $\delta \in (0,1)$, with probability at least $1-\delta$ over draws of $D \sim \nu^n$,*

$$\mathbb{E}_{\rho}[L(\boldsymbol{\theta})] \leq \mathbb{E}_{\rho}[\hat{L}(D,\boldsymbol{\theta})] + \inf_{\lambda\in[0,b)}\left\{\frac{KL(\rho|\pi)+\log\frac{n}{\delta}}{\lambda(n-1)} + \frac{\mathbb{E}_{\rho}[\Lambda_{\boldsymbol{\theta}}(\lambda)]}{\lambda}\right\}$$

*simultaneously for every $\rho \in \mathcal{M}_1(\boldsymbol{\Theta})$.*

*Proof.* For any posterior distribution $\rho \in \mathcal{M}_1(\boldsymbol{\Theta})$ and any positive $m < n$, consider $m\Lambda_{\rho}^{\star}(gen(\rho,D))$, where $gen(\rho,D) := \mathbb{E}_{\rho}[L(\boldsymbol{\theta})] - \mathbb{E}_{\rho}[\hat{L}(D,\boldsymbol{\theta})]$. The function $\Lambda_{\rho}^{\star}(\cdot)$ will play a role analogue to the convex comparator function in Rivasplata et al. (2020). Since $\sup_{\lambda}\mathbb{E}X_{\lambda} \leq \mathbb{E}\sup_{\lambda}X_{\lambda}$, it verifies that

$$m\Lambda_{\rho}^{\star}(gen(\rho,D)) \leq m\mathbb{E}_{\rho}[\Lambda_{\boldsymbol{\theta}}^{\star}(gen(\boldsymbol{\theta},D))]. \tag{17}$$

Applying Donsker-Varadhan's change of measure (Donsker and Varadhan, 1975) to the right-hand side of the inequality we obtain

$$m\Lambda_{\rho}^{\star}(gen(\rho,D)) \leq KL(\rho|\pi) + \log\mathbb{E}_{\pi}\left(e^{m\Lambda_{\boldsymbol{\theta}}^{\star}(gen(\boldsymbol{\theta},D))}\right). \tag{18}$$

We can now apply Markov's inequality to the random variable $\mathbb{E}_{\pi}\left(e^{m\Lambda_{\boldsymbol{\theta}}^{\star}(gen(\boldsymbol{\theta},D))}\right)$. Thus, with probability at least $1-\delta$,

$$m\Lambda_{\rho}^{\star}(gen(\rho,D)) \leq KL(\rho|\pi) + \log\frac{1}{\delta} + \log\mathbb{E}_{\nu^n}\mathbb{E}_{\pi}\left(e^{m\Lambda_{\boldsymbol{\theta}}^{\star}(gen(\boldsymbol{\theta},D))}\right). \tag{19}$$

Since $\pi$ is data-independent, we can swap both expectations using Fubini's theorem, so that we need to bound $\mathbb{E}_{\nu^n}\left(e^{m\Lambda_{\boldsymbol{\theta}}^{\star}(gen(\boldsymbol{\theta},D))}\right)$ for any fixed $\boldsymbol{\theta} \in \boldsymbol{\Theta}$. Here is where Lemma 6 comes into play: we have that for any $c > 0$,

$$\mathbb{P}_{D\sim\nu^n}\left(n\Lambda_{\boldsymbol{\theta}}^{\star}(gen(\boldsymbol{\theta},D))\geq\frac{n}{m}c\right) \leq \mathbb{P}_{X\sim\exp(1)}\left(X\geq\frac{n}{m}c\right).$$

Since $X \sim \exp(1)$, we get $kX \sim \exp(\frac{1}{k})$. Thus, multiplying by $\frac{m}{n}$,

$$\mathbb{P}_{D\sim\nu^n}\left(m\Lambda_{\boldsymbol{\theta}}^{\star}(gen(\boldsymbol{\theta},D))\geq c\right) \leq \mathbb{P}_{X\sim\exp(\frac{n}{m})}\left(X\geq c\right). \tag{20}$$

which in turn results in

$$\mathbb{P}_{\nu^n}\left(e^{m\Lambda_{\boldsymbol{\theta}}^{\star}(gen(\boldsymbol{\theta},D))} \geq t\right) \leq \mathbb{P}_{\exp\left(\frac{n}{m}\right)}\left(e^X \geq t\right) \tag{21}$$

for any $t \geq 1$. Finally, since $X \sim \exp(\frac{n}{m})$, we have $e^X \sim \text{Pareto}\left(\frac{n}{m}, 1\right)$. Thus, for any $t \geq 1$

$$\mathbb{P}_{\nu^n}\left(e^{m\Lambda_{\boldsymbol{\theta}}^{\star}(gen(\boldsymbol{\theta},D))} \geq t\right) \leq \mathbb{P}_{\text{Pareto}\left(\frac{n}{m},1\right)}\left(X \geq t\right). \tag{22}$$

Using that for any random variable $Z$ with support $\Omega \subseteq \mathbb{R}_+$, its expectation can be written as

$$\mathbb{E}[Z] = \int_{\Omega} P(Z \geq z)dz, \tag{23}$$

we obtein the desired bound:

$$\begin{aligned}
\mathbb{E}_{D\sim\nu^n}\left(e^{m\Lambda_{\boldsymbol{\theta}}^{\star}(gen(\boldsymbol{\theta},D))}\right) &= \int_1^{\infty} \mathbb{P}_{D\sim\nu^n}\left(e^{m\Lambda_{\boldsymbol{\theta}}^{\star}(gen(\boldsymbol{\theta},D))} \geq t\right)dt \\
&\leq \int_1^{\infty} \mathbb{P}_{X\sim\text{Pareto}\left(\frac{n}{m},1\right)}\left(X \geq t\right)dt \\
&= \mathbb{E}_{X\sim\text{Pareto}\left(\frac{n}{m},1\right)}(X) \\
&= \frac{\frac{n}{m}}{\frac{n}{m}-1} = \frac{n}{n-m}.
\end{aligned}$$

Observe how the condition $m < n$ is crucial because a Pareto$(1,1)$ has no finite mean. In conclusion, with probability at least $1 - \delta$ we have

$$m\Lambda_{\rho}^{\star}\big(gen(\rho,D)\big) \leq KL(\rho|\pi) + \log\frac{n}{n-m} + \log\frac{1}{\delta}. \tag{24}$$

Dividing by $m$, setting $m = n - 1$ and applying $(\Lambda_{\rho}^{\star})^{-1}(\cdot)$ in both sides concludes the proof. $\qquad\square$

**Corollary 8.** *Let $\ell$ be the $0 - 1$ loss and $\pi \in \mathcal{M}_1(\boldsymbol{\Theta})$ be any prior independent of $D$. Then, for any $\delta \in (0,1)$, with probability at least $1 - \delta$ over draws of $D \sim \nu^n$,*

$$kl\left(\mathbb{E}_{\rho}[\hat{L}(\boldsymbol{\theta},D)], \mathbb{E}_{\rho}[L(\boldsymbol{\theta})]\right) \leq \frac{KL(\rho|\pi) + \log\frac{n}{\delta}}{n-1},$$

*simultaneously for every $\rho \in \mathcal{M}_1(\boldsymbol{\Theta})$.*

*Proof.* We know that $\ell(\boldsymbol{\theta}, X) \sim \text{Bin}(L(\boldsymbol{\theta}))$, hence following the approach in Section 2.2 of Boucheron et al. (2013), we obtain

$$\Lambda_{\boldsymbol{\theta}}^{\star}(a) = kl\left(L(\boldsymbol{\theta}) - a|L(\boldsymbol{\theta})\right).$$

From the proof of Theorem 7 we have

$$\mathbb{E}_{\rho}[\Lambda_{\boldsymbol{\theta}}^{\star}(gen(\boldsymbol{\theta},D))] \leq \frac{KL(\rho|\pi) + \log\frac{n}{\delta}}{n-1}, \tag{25}$$

and the result follows from the convexity of $kl$ and Jensen's inequality. $\qquad\square$

**Proposition 12.** *If we fix some $\lambda > 0$, the bound in Theorem 11 can be minimized with respect to $\rho \in \mathcal{M}_1(\boldsymbol{\Theta})$. The optimal posterior is*

$$\rho^*(\boldsymbol{\theta}) \propto \pi(\boldsymbol{\theta})\exp\left\{-(n-1)\lambda\hat{L}(D,\boldsymbol{\theta}) - (n-1)\psi(\boldsymbol{\theta},\lambda)\right\}. \tag{12}$$

*Proof.* We can solve the constrained minimization problem using standard results from variational calculus —see Appendices D and E of Bishop (2006) for a succinct introduction—. We need to minimize the functional

$$\mathcal{B}_{\pi,\lambda}[\rho] := \mathbb{E}_{\rho}[\hat{L}(D,\boldsymbol{\theta})] + \frac{\mathbb{E}_{\rho}[\psi(\boldsymbol{\theta},\lambda)]}{\lambda} + \frac{KL(\rho|\pi) + \log\frac{n}{\delta}}{\lambda(n-1)} + \gamma\left(\int_{\Theta}\rho(\boldsymbol{\theta})d\theta - 1\right),$$

where $\gamma \geq 0$ is the Lagrange multiplier. For this purpose, we compute the functional derivative of $\mathcal{B}_{\pi,\lambda}[\rho]$ wrt $\rho$,

$$\frac{\delta \mathcal{B}_{\pi,\lambda}}{\delta \rho} = \hat{L}(D, \boldsymbol{\theta}) + \frac{\psi(\boldsymbol{\theta}, \lambda)}{\lambda} + \frac{1}{\lambda(n-1)} \left( \log \frac{\rho}{\pi} + 1 \right) + \gamma,$$

and find $\rho \in \mathcal{M}_1(\boldsymbol{\Theta})$ satisfying $\frac{\delta \mathcal{B}_{\pi,\lambda}}{\delta \rho} = 0$, which results in the desired $\rho^*$ after straightforward algebraic manipulations. $\qquad \square$

**Theorem 16.** *If Assumptions 2 and 3 are satisfied, then for any prior distribution $\pi \in \mathcal{M}_1(\boldsymbol{\Theta})$ independent of $D$ and any $\delta \in (0,1)$; with probability at least $1 - \delta$ over draws of $D \sim \nu^n$,*

$$\mathbb{E}_\rho[L(\boldsymbol{\theta})] \leq \mathbb{E}_\rho[\hat{L}(D, \boldsymbol{\theta})] + \sqrt{2C\mathbb{E}_\rho\left[\|\widehat{\nabla}_{\boldsymbol{x}}\ell\|_2^2\right] \cdot K(\rho, \pi, n, \delta) + \sqrt{2}CL \cdot K(\rho, \pi, n, \delta)^{\frac{3}{2}}}$$

*simultaneously for every $\rho \in \mathcal{M}_1(\boldsymbol{\Theta})$, where $K(\rho, \pi, n, \delta) := \frac{KL(\rho|\pi) + \log \frac{2n}{\delta}}{n-1}$.*

*Proof.* By Assumption 3, $\|\nabla_{\boldsymbol{x}}\ell(\boldsymbol{\theta}, \boldsymbol{x})\|_2^2 \leq L$ for any $\boldsymbol{\theta} \in \boldsymbol{\Theta}$ and any $\boldsymbol{x} \in \mathcal{X}$ —note that with this we can already obtain a bound that includes a model-dependent Lipschitz constant $L(\boldsymbol{\theta})$—. In particular, $\|\nabla_{\boldsymbol{x}}\ell(\boldsymbol{\theta}, X)\|_2^2$ is $\frac{L^2}{4}$-sub-Gaussian. Thus we can use Corollary 5.3 in Hellström et al. (2023) and obtain the following PAC-Bayes bound:

$$\mathbb{E}_\rho\left[\|\nabla_{\boldsymbol{x}}\ell\|_2^2\right] \leq \mathbb{E}_\rho\left[\|\widehat{\nabla}_{\boldsymbol{x}}\ell\|_2^2\right] + \frac{L}{\sqrt{2}}\sqrt{\frac{KL(\rho|\pi) + \log \frac{\sqrt{n}}{\delta_2}}{n-1}}, \qquad (26)$$

with probability at least $1 - \delta_2$.

Now taking $\delta_1 = \delta_2 = \frac{\delta}{2}$, the bound in Theorem 15 and the one in (26) hold simultaneously with probability at least $1 - \delta$. Finally, plugging (26) in Theorem 15 we obtain the desired result. $\qquad \square$

# B  PAC-Bayes bounds for losses with bounded CGF

**Corollary 19.** *Assume the loss is $\sigma^2$-sub-Gaussian. Let $\pi \in \mathcal{M}_1(\boldsymbol{\Theta})$ be any prior independent of $D$. Then, for any $\delta \in (0,1)$, with probability at least $1 - \delta$ over draws of $D \sim \nu^n$,*

$$\mathbb{E}_\rho[L(\boldsymbol{\theta})] \leq \mathbb{E}_\rho[\hat{L}(D, \boldsymbol{\theta})] + \sqrt{2\sigma^2 \frac{KL(\rho|\pi) + \log \frac{n}{\delta}}{n-1}},$$

*simultaneously for every $\rho \in \mathcal{M}_1(\boldsymbol{\Theta})$.*

*Proof.* Follows from the fact that $(\psi^*)^{-1}(s) = \sqrt{2\sigma^2 s}$ for $\sigma^2$-sub-Gaussian random variables (Boucheron et al., 2013, Section 2). $\qquad \square$

**Corollary 20.** *Assume the loss is $(\sigma^2, c)$-sub-gamma. Let $\pi \in \mathcal{M}_1(\boldsymbol{\Theta})$ be any prior independent of $D$. Then, for any $\delta \in (0,1)$, with probability at least $1 - \delta$ over draws of $D \sim \nu^n$,*

$$\mathbb{E}_\rho[L(\boldsymbol{\theta})] \leq \mathbb{E}_\rho[\hat{L}(D, \boldsymbol{\theta})] + \sqrt{2\sigma^2 \frac{KL(\rho|\pi) + \log \frac{n}{\delta}}{n-1}} + c\frac{KL(\rho|\pi) + \log \frac{n}{\delta}}{n-1},$$

*simultaneously for every $\rho \in \mathcal{M}_1(\boldsymbol{\Theta})$.*

*Proof.* Follows from the fact that $(\psi^*)^{-1}(s) = \sqrt{2\sigma^2 s} + cs$ for $(\sigma^2, c)$-sub-gamma random variables (Boucheron et al., 2013, Section 2). $\qquad \square$

# C   Experimental details

## C.1   Models and data

As for the experimental setting for Figure 1 and the verification of Assumption 2; we have used the small InceptionV3 architecture (Szegedy et al., 2016) used in Zhang et al. (2017), where the total number of parameters of the model is 1.814.106. We trained the model in the CIFAR10 dataset (Krizhevsky et al., 2009) with the default train/test split using SGD with momentum 0.9 and learning rate 0.01 with exponential decay of 0.95. All models are trained for 30.000 iterations of batches of size 200 or until the train loss is under 0.005. These settings are selected to ensure that the random label model converges to an interpolator. For $\ell_2$ regularization, the multiplicative factor is 0.01. We include a Jupyter Notebook in the Supplementary Material with the code for reproducing our experiments and figures.

From the definition of the log-loss, we have

$$\Lambda_{\boldsymbol{\theta}}(\lambda) = \ln \mathbb{E}_{\nu}\left[e^{\lambda(L(\boldsymbol{\theta})-\ell(\boldsymbol{y},\boldsymbol{x},\boldsymbol{\theta}))}\right] = \ln \mathbb{E}_{\nu}\left[p(\boldsymbol{y}|\boldsymbol{x},\boldsymbol{\theta})^{\lambda}\right] - \mathbb{E}_{\nu}[\ln p(\boldsymbol{y}|\boldsymbol{x},\boldsymbol{\theta})^{\lambda}],$$

and we estimate the expectations using the test data:

$$\Lambda_{\boldsymbol{\theta}}(\lambda) \approx \ln \left(\frac{1}{M}\sum_{(\boldsymbol{x},\boldsymbol{y})\in D^{test}} p(\boldsymbol{y}|\boldsymbol{x},\boldsymbol{\theta})^{\lambda}\right) - \frac{1}{M}\sum_{(\boldsymbol{x},\boldsymbol{y})\in D^{test}} \ln p(\boldsymbol{y}|\boldsymbol{x},\boldsymbol{\theta})^{\lambda}.$$

We also estimated $\mathbb{V}(\ell)$ and $\|\nabla_x\ell\|_2$ using $D^{test}$.

## C.2   Experimental backing for Assumption 2

We now experimentally evaluate Assumption 2 in the same setup as above. We need to show that there is certain $C > 0$ such that for any $\lambda > 0$, we have

$$\Lambda_{\boldsymbol{\theta}}(\lambda) \le \frac{C}{2}\lambda^2\|\nabla_{\boldsymbol{x}}\ell\|_2^2,$$

for every $\boldsymbol{\theta} \in \boldsymbol{\Theta}$. Since in this case each $\boldsymbol{\theta}$ represents a configuration of weights in the InceptionV3 architecture, we cannot strictly verify the assumption for every $\boldsymbol{\theta}$, however, we provide empirical backing for its feasibility.

We estimate $\Lambda_{\boldsymbol{\theta}}(\lambda)$ and $\|\nabla_{\boldsymbol{x}}\ell\|_2^2$ with test data and plot $\frac{\Lambda_{\boldsymbol{\theta}}(\lambda)}{0.5\lambda^2\|\nabla_{\boldsymbol{x}}\ell\|_2^2}$ for the three different models introduced in Figure 1 (in the case of Zero the inequality trivially holds).

We plot the results in Figure 2, showing that the quantity of interest decreases with $\lambda$ and that it is reasonable to expect Assumption 2 holding with $C < 1$. Observe that by a simple call to L'Hopital's rule, one can see that $\lim_{x\to 0^+} \frac{\Lambda_{\boldsymbol{\theta}}(\lambda)}{0.5\lambda^2\|\nabla_{\boldsymbol{x}}\ell\|_2^2} = \frac{\mathbb{V}(\ell)}{\|\nabla_{\boldsymbol{x}}\ell\|_2^2}$.

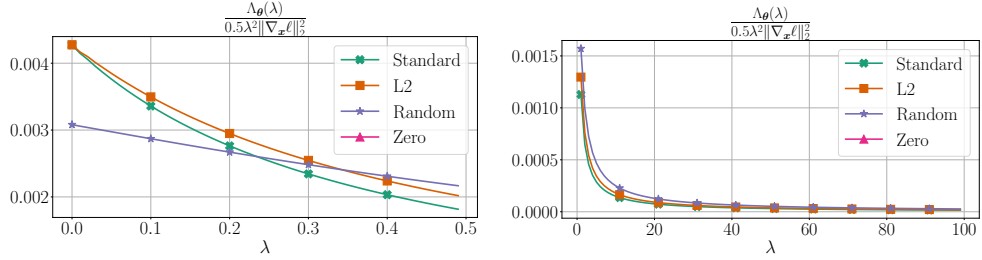

Figure 2: The plots of $\frac{\Lambda_{\boldsymbol{\theta}}(\lambda)}{0.5\lambda^2\|\nabla_{\boldsymbol{x}}\ell\|_2^2}$ in two different scales. In the case of these models, Assumption 2 holds with $C \approx 0.005$.

