# OpenReview forum: "PAC-Bayes-Chernoff bounds for unbounded losses"
_NeurIPS.cc/2024/Conference — NeurIPS 2024 poster_

### Official Review · Reviewer_Kbx9 · 2024-07-01

**Soundness:** 3
**Presentation:** 3
**Contribution:** 3
**Rating:** 7
**Confidence:** 4

**Summary:**

The authors propose a new oracle PAC-Bayesian bound that has two main features: it is valid for unbounded losses (or at least under assumptions weaker than bounded loss) and it allows for an exact optimization of the free parameter $\lambda$ appearing in most PAC-Bayesian bound, with only the cost of a penalty that is logarithmic in the number of data points. Based on this new bound, the authors first recover existing bounds and improve over the prior art by introducing model-dependent assumptions in the generalization bounds. They also make the link with regularization techniques. In particular, they obtain new bounds based on input-gradients by combining their theory with log-Sobolev-type inequalities.

**Strengths:**

- The ability of exactly optimizing the free parameter $\lambda$ in an oracle PAC-Bayesian bound is a nice contribution. It may be useful in several settings.

 - The proofs of the main results rely on quite general bounded CGF assumptions, which are weaker than the usual bounded loss assumption and allow for model-dependent assumptions.

 - Instead of usual exponential terms that are averaged over the prior distribution, the authors provide PAC-Bayesian bounds with a stronger dependence on the posterior distribution, hence leveraging the concentration properties of each individual model. Beyond tightening the generalization bounds, this could strengthen the practical relevance of PAC-Bayesian theory.

**Weaknesses:**

- It should be made more clear how the main results compare to existing results, especially the ones relying on grids and union bounds, as well as other PAC-Bayesian bounds that do not have neither a $\log(n)$ penalty nor free parameters for subgaussian (hence unbounded) losses. These existing results should be written explicitly and compared with the new bounds.

 - Adding a few examples of situations where the new bounds clearly outperform the existing ones would enhance the paper.

 - Additional technical background on the generalized inverse and its main properties would help a lot the readability of the paper, as it is the main technical ingredient.

 - The log-Sobolev inequalities mentioned in the last section seem to be stronger than the usual log-Sobolev inequality. Some clarification should be added (see the questions).

 - There might be a mistake in the statement of Theorem 1. Shouldn't it be $1/\lambda$ instead of $1/(\lambda n)$? In Theorem 3 of [Germain et al., 2016], the result is stated with $1/\lambda$ instead of $1/(\lambda n)$.

**Questions:**

- Do you know any explicit example where your oracle bound yields a better optimization of $\lambda$ than the best existing results under the same assumptions (obtained either by union bound arguments or other techniques)?

 - In Lemma 18, you prove that $\Lambda_\theta^* (a) < \infty$ on $[0,L(\theta))$, but then in the proofs of Lemma 6 and theorem 7, you apply $\Lambda_\theta^*$ on $gen(\theta, D)$, which may take negative values. Why is that justified?

 - Corollary 13: the result would be even stronger if the expectation over the posterior was outside of the square root. Do you think it is possible to obtain such a result?

 - After Assumption 2, line 279, you claim that the log-Sobolev inequality state in Assumption 2 holds for several well-known distributions, such as the Gaussian distribution. However, in the reference [50] you give for this result, a much weaker inequality appears. Indeed, the log-Sobolev inequality would be $\Lambda_\theta (\lambda) \leq \frac{C}{2} \lambda^2 L(\theta)^2$, where $L(\theta)$ is the Lipschitz constant of $x \longmapsto \nabla_x \ell(x, \theta)$. Can you prove that Assumption 2 indeed holds for the Gaussian distribution, or give a reference?

 - Line 250: where exactly is this result proven in the cited paper? Can you be more precise about how this result is obtained?

**Minor remarks / questions:**

 - Line 26: a $\to$ an.

 - Line 17: The notation $\mathcal{M}_1(\Theta)$ should be introduced.

 - Line 131: space permits to use the long notations in most cases, but it is not done, while it would improve the readability of the paper.

 - Line 243: regularizaton $\to$ regularization

 - Line 244: based in $\to$ based on.

 - Assumption 1: if $\ell$ is $M$-Lipschitz in $\theta$, then $\Vert \nabla_\theta \ell (x,\theta)\Vert_2^2$ is bounded by $M^2$, not $M$.

 - We see in Appendix A that the fact that $\ell\geq 0$ plays an important role in the proofs. Does your theory hold if $\ell$ is not lower bounded?

 - In the definition of $\Lambda_\theta^*$, could the parameter $b$ depend on $\theta$? Could that cause any issue in your proofs?

**Limitations:**

Most of the limitations were discussed in the paper.

The paper has no societal or ethical impact.

---

> ### Author Rebuttal · Authors · 2024-08-07
>
> We are grateful for your detailed feedback, it will definitely improve the quality of our paper.
>
> > It should be made more clear how the main results compare to existing results [...]
>
> Thank you for the suggestion. You are correct that our discussion is somewhat incomplete and scattered throughout the paper. We will add a new subsection in the revised version to compare our results with existing findings. Please note that we compare our bound with Theorem 14 of [1] in lines 189-194, demonstrating that our bound is tighter for large KL divergences. Additionally, [1] offers state-of-the-art bounds for unbounded losses and enhances the grid+union bound method.
>
> In relation to your comment about  *PAC-Bayesian bounds that do not have neither a $\log(n)$ penalty nor free parameters for sub-Gaussian (hence unbounded) losses*, we understand you refer to sub-Gaussian bounds with $\log(1/\delta)$ dependency and no free parameters. As far as we know, most of those bounds, such as Corollary 4 in [2], simply hold for a fixed certain $\lambda$, which usually result in looser bounds than those with a $\log(n)$ penalty and optimized $\lambda$.
>
>
>
> > Adding a few examples of situations where the new bounds clearly outperform the existing ones would enhance the paper.
>
> Good suggestion. Probably, the most clear example is the one given in our discussion of Corollary 13 we observe how this bound outperforms standard sub-Gaussian bounds because of the model-dependent proxy variance improves upon the worst-case proxy variance. This case can be extended for sub-gamma random variables and the like. We will better highlight these cases in the new version of the paper. Furthermore, we will also highlight more how Theorems 15 and 16 improve those in [3] for qualitative reasons (actually relating posterior performance with gradient norms and optimization of $\lambda$).
>
>
> > Additional technical background on the generalized inverse and its main properties would help a lot the readability of the paper, as it is the main technical ingredient.
>
> This is also a reasonable suggestion, we'll add the definitions and lemmas used in the proofs in a separate Appendix Section.
>
>
> > There might be a mistake in the statement of Theorem 1. Shouldn't it be $1/\lambda$ instead of $1/(\lambda n$)? [...]
>
> This is just a matter of the parametrization you choose, see for example Theorem 1 in [4] and the discussion in Footnote 3. We used $\lambda n$ to emphasize the dependence on the dataset size $n$.
>
> > Do you know any explicit example where your oracle bound yields a better optimization of $\lambda$ [...] ?
>
> As discussed above, our Corollary 9 provides better optimization than the method of Theorem 14 in [1] when the KL term is large. But, again, when making these comparisons we are forced to discard one of the main contributions of our bounds, that is, the model-dependent bounding term.
>
>
> > In Lemma 18 [...], which may take negative values. Why is that justified?
>
> Observe that $\Lambda^*_\theta(a):=\sup_{\lambda>0} \{\lambda a - \Lambda_\theta(\lambda)\}$, hence $\Lambda^*_\theta(a)=0$ for negative values of $a$ because $\Lambda_\theta(\lambda)$ is always positive. See our discussion with Reviewer MaqK for more context.
>
> > Corollary 13: the result would be even stronger if the expectation over the posterior was outside of the square root. Do you think it is possible to obtain such a result?
>
> Not with our approach because the square root is a consequence of optimizing $\lambda$ and the optimization depends on $\rho$ and $\sigma(\theta)^2$.
>
>
> > After Assumption 2, line 279, you claim that the log-Sobolev inequality state in Assumption 2 [...]
>
> Our Assumption 2 follows from Corollary 9 in [50] using $\phi=-\log f$ and $f=\exp(-\lambda \ell)$. Note that Corollary 9 in [50] directly involves the expected norm of the gradients. We will include a formal result in the appendix for completeness. The Gaussian case is a particular case of Corollary 9.
>
> > Line 250: where exactly is this result proven in the cited paper? Can you be more precise about how this result is obtained?
>
> This is shown in the proof of the Proposition 5.2 of [37], where we only use that the model $\theta_0$ with all parameters set to zero has zero variance. For completeness, we will include the complete proof in the camera-ready version.
>
> > We see in Appendix A that the fact that $\ell\geq 0$
> plays an important role in the proofs. Does your theory hold if $\ell$ is not lower bounded?
>
> See the answer to Reviewer MaqK related to the sub-exponential assumption. In sort, if $\ell$ is not lower bounded we would need to make a sub-exponential assumption. But, in ML, having not lower-bounded lossess hardly make sense.
>
>
> > In the definition of $\Lambda^*_\theta$, could the parameter $b$ depend on $\theta$? Could that cause any issue in your proofs?
>
> Observe that in the Appendix A (Line 515) we show that $b=\infty$ for all $\theta$ when $L(\theta)$ is finite, which is the standard case. In more general cases, yes, it can happen and this would affect the definition of $\Lambda^*_\rho$ in Line 150. The interesting thing is that, even when $\Lambda_\theta$ are finite in $[0,b_\theta)$ for different $b_\theta$'s, $E_\rho[\Lambda_\theta(\lambda)]$ is also finite in some interval $[0,c)$, where $c$ depends on the $b_\theta$, and everything works normally.
>
> [1] Rodríguez-Gálvez, B. et al. (2024). More PAC-Bayes bounds: From bounded losses, to losses with general tail behaviors, to anytime validity. Journal of Machine Learning Research.
>
> [2] Germain, P. et al. (2016). PAC-Bayesian theory meets Bayesian inference. Advances in Neural Information Processing Systems, 29.
>
> [3] Gat, I. et al. (2022). On the importance of gradient norm in PAC-Bayesian bounds. Advances in Neural Information Processing Systems, 35.
>
> [4] Masegosa, A. (2020). Learning under model misspecification: Applications to variational and ensemble methods. Advances in Neural Information Processing Systems, 33.

---

> > ### Comment · Reviewer_Kbx9 · 2024-08-09
> > **Thank you, and a few additional questions**
> >
> > I want to thank the authors for taking the time to appropriately address my concerns.
> >
> > I just have a two additional minor questions that I would like to ask.
> >
> >  - Regarding Theorem 1, while this is only a minor issue, I respectfully disagree. If you use this parameterization, $\lambda$ should be replaced by $\lambda n$ inside $f_{\pi,\lambda}$. Please correct me if I am wrong.
> >
> >  - Regarding the log-Sobolev inequality, my concern was that there is no expectation on the norm of the gradient in Assumption 2, but maybe it is still a consequence of Corollary 9 in [50]?
> >
> >  Finally, regarding bounds for sub-Gaussian losses without log(n) penalty, I was referring to Theorem 2.1 in [1]. Please note that I think that your result is significant as it allows for the optimization of $\lambda$ in a lot of different settings. I don't see this $\log(n)$ difference as an issue.
> >
> > Reference:
> > [1] Benjamin Dupuis and Umut Şimşekli. Generalization bounds for heavy-tailed sdes through the fractional Fokker-Planck equation, 2024.

---

> > > ### Author Response · Authors · 2024-08-10
> > >
> > > Thanks a lot for your quick response, your really good feedback and your very detailed comments. We really appreciate all of this.
> > >
> > > > Regarding Theorem 1 [...]
> > >
> > > Sorry. You are totally right! There is indeed a typo in  Theorem 1. We omitted the $n\lambda$ term in $f_{\pi,\lambda}$. Of course, it can be reparametrized using only $\lambda$. But we had an error there. Good catch!. In any case, this does not affect any of the discussions related to the theorem.
> > >
> > > > Regarding the log-Sobolev inequality, [...]
> > >
> > > This is a misunderstanding with the notation. We will update the notation to avoid confusion. Please look at the definition given in Line 270.  There, we define:
> > > $$\|\nabla_x\ell\|^2_2 := E_\nu [\|\nabla_x \ell(x,\theta)\|_2^2 ]$$
> > > Assumption 2 does really involve an expectation on the norm of the gradient.
> > >
> > > > Finally, regarding bounds for sub-Gaussian losses without log(n) penalty, [...]
> > >
> > > Again, there is a misunderstanding. We referred to the results given in Corollary 2 in [36] or Corollary 19 in our paper. We were not aware of Theorem 2.1 in [1], it is a very recent paper (june 2024). But this is indeed a stronger result for sub-Gaussian random variables because, as you mention, does not include any log n term! Thanks for the reference. We will will include it in the updated version of the paper

---

> > > > ### Comment · Reviewer_Kbx9 · 2024-08-10
> > > > **Thanks**
> > > >
> > > > Thank you for your response.
> > > >
> > > > Indeed, I had misunderstood the notation for the gradient norm. A change of notation could improve the quality of the paper.
> > > >
> > > > All my concerns have been appropriately addressed, I will increase my score to 7 (accept).

---

### Official Review · Reviewer_JzAv · 2024-07-09

**Soundness:** 3
**Presentation:** 3
**Contribution:** 3
**Rating:** 7
**Confidence:** 3

**Summary:**

This paper gives novel PAC-Bayes oracle bounds using the Cramer transform's basic properties under a bounded exponential moment condition.

The benefit of using the Cramer transform is that the bound allows exact optimization of the free parameter $\lambda$ incurring in a $\log n$ penalty without resorting to union-bound approaches, typically used in the related literature.

Then, by considering a model-dependent bounded exponential moment condition, the bound would be tightened. In this case, the posterior distribution can be optimized, which results in optimal distributions beyond Gibbs’ posterior.

By applications to generalized sub-Gaussian losses, norm-based regularization, and log-Sobolev inequalities, tighter and novel bounds are provided.

**Strengths:**

The introduction is written very clearly.

Many insights concerning the PAC-Bayes are given: exact optimization of the free parameter λ, designing of better posteriors, and tighter bounds, etc.

**Weaknesses:**

I don't understand the author's claim in the first paragraph of the Section Limitations and future work: An apparent limitation of our approach is that the we are implicitly assuming that ℓ(θ, X) is light tailed (equivalently, sub-exponential), as in every Cramér-Chernoff bound. This is only partially true.

Lemma 6 gives an upper bound of sub-exponential random variables. In my opinion, the results of this paper seem to only apply to sub exponential losses. Can the author provide more explanations？

**Questions:**

Can Lemma 6 be bounded by a subgaussian random variable for subgaussian loss. Will there be better generalization results in this situation?

**Limitations:**

Yes

---

> ### Author Rebuttal · Authors · 2024-08-07
>
> We sincerely thank you for your review, we are happy that you appreciated the clarity and the insights of our work. We address your questions below.
>
>
> > I don't understand the author's claim in the first paragraph of the Section Limitations and future work: An apparent limitation of our approach is that the we are implicitly assuming that $\ell(\theta, X)$ is light tailed (equivalently, sub-exponential), as in every Cramér-Chernoff bound. This is only partially true.
> Lemma 6 gives an upper bound of sub-exponential random variables. In my opinion, the results of this paper seem to only apply to sub exponential losses. Can the author provide more explanations?
>
>
> This is a fair question that will be clarified in more detail using the extra page available for the camera-ready version.
>
>
> Since we assume the loss function $\ell$ is positive, the random variable $L(\theta) - \ell(\theta, X)$ is upper-bounded by $L(\theta)$. Therefore, we don't need to make assumptions about the right tail of $L(\theta) - \ell(\theta, X)$, other than ensuring that the expected value $L(\theta)$ is finite. However, if we were interested in bounding $|L(\theta) - \ell(\theta, X)|$, we would also need to control the left tail of $L(\theta) - \ell(\theta, X)$. The left tail can be heavy-tailed, which would require the sub-exponential assumption to manage effectively.
>
> Since we assume the loss function $\ell$ is positive, the random variables we are interested in upper-bounding, $L(\theta) - \ell(\theta, X)$, are all bounded by $L(\theta)$, hence we don't need to assume anything about the right tail of $L(\theta) - \ell(\theta, X)$ (except for the expected value $L(\theta)$ being finite, of course). However, if we were interested in bounding $|L(\theta) - \ell(\theta, X)|$, we would need to control the left tail of $L(\theta) - \ell(\theta, X)$ too, which can be heavy-tailed and would need the sub-exponential assumption to work.
>
> The fact that we only care on the right tail of $L(\theta) - \ell(\theta, X)$ is also implicit on the fact that the CGF (or the CGF bounding function) is only defined for $\lambda>0$. This fact is explicited, for example, in Definition 10 of [1] and the discussion below. The discussion on Section 2.2 of Chapter 2 in [2] is also clarifying.
>
> Most PAC-Bayes bounds only care about upper bounds on $L(\theta) - \ell(\theta, X)$ because in practice the empirical risk is usually much smaller than the actual one and because PAC-Bayesian learning works by minimizing upper bounds on $L(\theta) - \ell(\theta, X)$. If the loss function was not bounded from below or we cared about bounds on $|L(\theta) - \ell(\theta, X)|$, we would need sub-exponentiality. This is what we tried to express in Limitations and future work (Section 6), but maybe our explanations were too rushed. As mentioned before, we will provide a more detailed clarification in the camera-ready version to ensure better understanding.
>
>
> > Can Lemma 6 be bounded by a subgaussian random variable for subgaussian loss. Will there be better generalization results in this situation?
>
> If it can be done it is not straightforward, because Lemma 6 relies in $\Lambda^*$ cancelling out with $(\Lambda^*)^{-1}$, hence resulting in the same $\mathbf{P}_{exp(1)}(X\geq c)$ bound. However, tightening this result is an important line of research for future work because the penalty term in our bounds depends on Lemma 6.
>
>
> [1] Rodríguez-Gálvez, B., Thobaben, R., Skoglund, M. (2024). More PAC-Bayes bounds: From bounded losses, to losses with general tail behaviors, to anytime validity. Journal of Machine Learning Research, 25(110), 1-43.
>
> [2] Boucheron, S., Lugosi, G., Massart, P. (2013). Concentration inequalities: a non asymptotic theory of independence.

---

> > ### Comment · Reviewer_JzAv · 2024-08-14
> >
> > Thanks for the response. I maintain my score.

---

### Official Review · Reviewer_MaqK · 2024-07-12

**Soundness:** 4
**Presentation:** 4
**Contribution:** 4
**Rating:** 7
**Confidence:** 3

**Summary:**

The authors present a novel PAC-Bayes bound tailored for unbounded losses, akin to a PAC-Bayes version of the Cramér-Chernoff inequality. The provided bound allows exact optimization of the free parameter across various PAC-Bayes bounds, and leads to more informative and tighter bounds by incorporating "model-dependent" terms, such as gradient norms.

**Strengths:**

This is a strong paper, that addresses important points in PAC-Bayes. It is clear, well-written, theoretically sound and pleasant to read.

**Weaknesses:**

I really enjoyed the paper, and only have a few points:
- The only fully empirical bound is Theorem 16. However, the Lipschitz constant L is unknown in practice and has to be estimated. Does that affect significantly the tightness of the bound, as well as its minimization? Same question for Constant C.
- I'm a bit disappointed the minimization of this bound has not been addressed here. Is that due to a computational difficulty or simply left for future work?
- Could the authors develop on the behavior of the optimal posteriors?

**Questions:**

See above

**Limitations:**

See above

---

> ### Author Rebuttal · Authors · 2024-08-07
>
> Thank you very much for your review, we are happy to see that you appreciated our contributions and had a pleasant reading. We address your doubts below:
>
>
> > The only fully empirical bound is Theorem 16. However, the Lipschitz constant L is unknown in practice and has to be estimated. Does that affect significantly the tightness of the bound, as well as its minimization? Same question for Constant C.
>
> Very good question. Yes, computing both the Lipchitz constant and the constant C will probably have an impact in the tightness of the bounds. For the former, specific approaches has been devised [1] and it is an active area of research. Computing the constant C would be challenging too.
>
>
>
> > I'm a bit disappointed the minimization of this bound has not been addressed here. Is that due to a computational difficulty or simply left for future work?
>
> We agree that training models by PAC-Bayes bound minimization is the next natural step for our work, and this is definitely under our radar for future work. However, these practical procedures contain several technical challenges, which we cannot address in the limited space we have. These challenges include simultaneously optimizing $\lambda$ and $\rho$, which would require adapting the work of [2], and the adaptation of either *PAC-Bayes with backprop* [3] or variational inference methods to our bounds.
> As we said, this is promising but highly non-trivial work we are planning to carry out in future works.
>
>
> > Could the authors develop on the behavior of the optimal posteriors?
>
> If we look at the optimal posterior in Proposition 12, we can see that in cases where the model-dependent bounding term $\psi(\theta,\lambda)$ is independent from the data, the posterior can be interpreted as a standard Gibbs posterior where the $\exp(-(n-1)\psi(\theta,\lambda))$ term is absorbed by the prior. In the other case there is not much we can say at the moment, that is why we talk about MAP estimates in the paper: because the regularizing effect of $\psi(\theta,\lambda)$ is easier to interpret there. In any case, further study of these posteriors is an exciting line of future work.
>
>
> [1] Fazlyab, M., Robey, A., Hassani, H., Morari, M., Pappas, G. (2019). Efficient and accurate estimation of Lipschitz constants for deep neural networks. Advances in Neural Information Processing Systems, 32.
>
> [2] Thiemann, N., Igel, C., Wintenberger, O., Seldin, Y. (2017, October). A strongly quasiconvex PAC-Bayesian bound. In International Conference on Algorithmic Learning Theory (pp. 466-492). PMLR.
>
> [3] Rivasplata, O., Tankasali, V. M., Szepesvari, C. (2019). PAC-Bayes with backprop. arXiv preprint arXiv:1908.07380.

---

### Official Review · Reviewer_XnKo · 2024-07-12

**Soundness:** 3
**Presentation:** 3
**Contribution:** 3
**Rating:** 6
**Confidence:** 2

**Summary:**

The paper presents a PAC-Bayes bound for the unbounded loss setting, improving on some of the main drawbacks of previous work on such bounds. The first such drawback discussed is the dependence of the tightness of the such bounds on a priori chosen free parameters, something which can usually only be partially circumvented by union bounding over a grid of free parameters. The second is the uniform control of the cumulant generating function of the loss across a model class. The paper show how the the introduced bound eliminates the need for approximate optimization over the free parameters (and the concomitant union bounding procedure), and how show the framework leading to the main theorem can be extended, exploiting model-specific bounding of the GCF.

**Strengths:**

The paper introduces a novel PAC-Bayes bound in the challenging setting of unbounded loss, and motives the contribution with clear discussion of the issues with previous PAC-Bayes bounds for unbounded loss functions. The numerical example of Figure 1 is a nice touch, showcasing how uniformly bounding the cumulative generating function of different models really can be a significant source of looseness in realistic settings. The paper is generally very well-written.

**Weaknesses:**

My main concern would be the extent to which this work will be interesting to this particular community. The technical contribution seems both solid and potentially useful, but it may be a better fit in a more specific venue. I am not well-versed enough in the line of work to which this paper belongs to give meaningful technical critiques.

**Questions:**

Is there any particular reason to use the phrasing "$\pi$ independent of $D$" (e.g. line 34)? I know this means that $\pi$ is chosen independent of the training sample (which allows for a Fubini theorem application), but find the phrase odd given that $P$ is usually chosen with reference to some features of the data distribution.

**Limitations:**

Yes

---

> ### Author Rebuttal · Authors · 2024-08-07
>
> Thank you very much for your review, we are happy to see that you appreciated the clarity of our writing and the motivation for our contribution. As for the weaknesses and questions, we address them individually below.
>
>
> > My main concern would be the extent to which this work will be interesting to this particular community. The technical contribution seems both solid and potentially useful, but it may be a better fit in a more specific venue. I am not well-versed enough in the line of work to which this paper belongs to give meaningful technical critiques.
>
>
> There is a long history of theoretically inclined impactful contributions to the field of learning theory and PAC-Bayes in NeurIPS (see for example, [2,3,4]). In fact, the first appearances of PAC-Bayesian learning theory made their appearance in the NeurIPS conference with [2] in 2002; along with more modern studies such as [4] from 2022. Furthermore, a quick look at how many works in the NeurIPS website include the words "PAC-Bayes" in the title or abstract show more than 1500 results in Google (site:https://neurips.cc PAC-Bayes). Hence we believe the NeurIPS community is an appropriate audience for our contribution.
>
>
> > Is there any particular reason to use the phrasing "$\pi$ independent of $D$" (e.g. line 34)? I know this means that is chosen independent of the training sample (which allows for a Fubini theorem application), but find the phrase odd given that $\pi$ is usually chosen with reference to some features of the data distribution.
>
> In this paper, we consider the standard approach in PAC-Bayesian bounds of using data-independent priors (see, for example, [1]). These priors can be selected using a priori features of the data distribution, such as the standard assumptions on the tails of the distribution (see the example at the discussion in Section 6 of [3]). But as the reviewer points, it is possible to use data-dependent priors, for instance, by splitting the available training data into two independent sets and using the first to construct the data-dependent prior (see Sections 4.3 and 9.3 in [5]).
>
>
> [1] Rodríguez-Gálvez, B., Thobaben, R., Skoglund, M. (2024). More PAC-Bayes bounds: From bounded losses, to losses with general tail behaviors, to anytime validity. Journal of Machine Learning Research, 25(110), 1-43.
>
> [2] Langford, J., Shawe-Taylor, J. (2002). PAC-Bayes and margins. Advances in Neural Information Processing Systems, 15.
>
> [3] Germain, P., Bach, F., Lacoste, A., Lacoste-Julien, S. (2016). PAC-Bayesian theory meets Bayesian inference. Advances in Neural Information Processing Systems, 29.
>
> [4] Haddouche, M., Guedj, B. (2022). Online PAC-Bayes learning. Advances in Neural Information Processing Systems, 35, 25725-25738.
>
> [5] Sanae Lotfi, Pavel Izmailov, Gregory Benton, Micah Goldblum, and Andrew Gordon Wilson. Bayesian model selection, the marginal likelihood, and generalization. In International Conference on Machine Learning, pages 14223–14247. PMLR, 2022.

---

> > ### Comment · Reviewer_XnKo · 2024-08-09
> >
> > Thanks for the response - I retain my previous score.

---

### Author Rebuttal · Authors · 2024-08-07

We thank all four reviewers for their helpful questions and suggestions. We are happy to see that there is certain consensus on the clarity, the soundness and the contributions of our work. We hope we clarified your doubts, and we are willing to implement the suggested changes on the camera-ready version.

---

### Decision · Program_Chairs · 2024-09-25

**Decision:**

Accept (poster)

**Comment:**

Classical PAC Bayes theorem holds for bounded losses and mainly rely on concentration inequality techniques for bounded random variables. In this paper PAC bayes theorem is extended to unbounded losses by considering the tail behavior of the losses. Specifically, the authors seem to combine PAC Bayesian approach with the log-sobolev style tools for proving concentration inequalities. This yielded to the authors a neat PAC Bayesian extension that could apply to unbounded losses as long as the Cramer transform of the losses are well behaved. This is a neat result worth publishing.